# Keratins as an Inflammation Trigger Point in Epidermolysis Bullosa Simplex

**DOI:** 10.3390/ijms222212446

**Published:** 2021-11-18

**Authors:** Nadezhda A. Evtushenko, Arkadii K. Beilin, Anastasiya V. Kosykh, Ekaterina A. Vorotelyak, Nadya G. Gurskaya

**Affiliations:** 1Center for Precision Genome Editing and Genetic Technologies for Biomedicine, Pirogov Russian National Research Medical University, Ostrovityanova 1, 117997 Moscow, Russia; hopeevt@gmail.com (N.A.E.); arkadii.beilin@gmail.com (A.K.B.); avkosyh@gmail.com (A.V.K.); 2Koltzov Institute of Developmental Biology of Russian Academy of Sciences, Vavilova 26, 119334 Moscow, Russia; vorotelyak@yandex.ru; 3Shemyakin-Ovchinnikov Institute of Bioorganic Chemistry, Miklukho-Maklaya 16/10, 117997 Moscow, Russia

**Keywords:** epidermolysis bullosa simplex, skin, blistering, epidermis, keratinocyte, basal layer, keratin, mutation, aggregation, stress, phosphorylation, injury, inflammation, wound healing, proinflammatory cascade, chemokine, cytokine

## Abstract

Epidermolysis bullosa simplex (EBS) is a group of inherited keratinopathies that, in most cases, arise due to mutations in keratins and lead to intraepidermal ruptures. The cellular pathology of most EBS subtypes is associated with the fragility of the intermediate filament network, cytolysis of the basal layer of the epidermis, or attenuation of hemidesmosomal/desmosomal components. Mutations in keratins 5/14 or in other genes that encode associated proteins induce structural disarrangements of different strengths depending on their locations in the genes. Keratin aggregates display impaired dynamics of assembly and diminished solubility and appear to be the trigger for endoplasmic reticulum (ER) stress upon being phosphorylated by MAPKs. Global changes in cellular signaling mainly occur in cases of severe dominant EBS mutations. The spectrum of changes initiated by phosphorylation includes the inhibition of proteasome degradation, TNF-α signaling activation, deregulated proliferation, abnormal cell migration, and impaired adherence of keratinocytes. ER stress also leads to the release of proinflammatory danger-associated molecular pattern (DAMP) molecules, which enhance avalanche-like inflammation. Many instances of positive feedback in the course of cellular stress and the development of sterile inflammation led to systemic chronic inflammation in EBS. This highlights the role of keratin in the maintenance of epidermal and immune homeostasis.

## 1. Introduction

The mechanical properties of the skin are the result of the combination of different components acting inside and outside the skin cells. They include the cytoskeletal network, adhesion complexes, and ECM. All these components function depending on specialized fine-tuned cell-signaling pathways. Keratins are the main structural elements of the epidermis, forming an extensive cytoskeleton network in keratinocytes (KCs), providing mechanical stiffness, and protecting against environmental influences [1]. Epidermolysis bullosa simplex (EBS) is a well-known genodermatosis, associated with the disruption of the intermediate filament network or hemidesmosomal/desmosomal components [2]. According to data from the National Registry of EB, EB simplex (EBS) is one of the three main types of inherited epidermolysis bullosa (EB), being the most widespread variant (70%) of EB. The EBS group, in turn, includes several subtypes of the disease, distinguished by the severity of the manifestations and by the genes affected (Appendix A). The pathological patterns of most EBS subtypes include skin fragility, with general or localized blistering in stratified epithelia. Patients with EBS can be affected severely or mildly, only cutaneously or with multisystem involvement. The development of plantar keratoderma is typical. The blisters and erosions (clustered in severe forms of EBS) heal without scars. This specific feature distinguishes EBS from other EB. The availability of skin tissue for sampling biopsies and advances in culture methods make EBS KCs a suitable model for the investigation of disease mechanisms. Patient-specific primary cell lines and immortalized cell lines became classical models for the evaluation of the impact of EBS mutations and the consequences of the disruption of the keratin network on stress sensitivity, KC migration, and the adhesion properties of the cell matrix [3]. Animal models of EBS (mainly murine) are also useful for specifically analyzing the mechanism of disease pathogenesis, as well as testing plausible drug therapy and genome-engineering approaches [4,5,6].

Mutations affecting protein folding were shown to induce endoplasmic reticulum (ER) stress and to activate stress-mediated phosphorylation and the proinflammatory induction of apoptosis [7]. The role of aggregation and fragmentation in keratin’s cytoskeleton was firstly uncovered as the cause of EBS’ molecular pathogenesis [8]. However, the direct manifestation of skin abnormalities in EBS, such as blisters and erosions, is driven by chronic inflammation and the pre-activation of the immune system due to the production of chemokines and cytokines [9,10,11]. The unfolded protein response (UPR) in a cell is activated by stress in the endoplasmic reticulum (ER) and initiates stress-induced homeostasis in the epidermis with chronic sterile inflammation, as in cases of other skin inflammatory diseases [12]. Several anomalies have been identified in gene cascades involved in cell proliferation, which are elements of the bone morphogenetic protein (BMP) signaling pathway, fatty acid metabolism, and retinoic acid signaling, as well as genes involved in the regulation of keratinization [13,14]. Additionally, microbial colonization aggravates the course of EBS. We consider possible causes of the enhancement of microbial infection and point to the possible involvement of keratin disruption in thymus medullary epithelial cells [15,16].

## 2. Several Aspects of Epidermal Keratinocyte Growth, Proliferation, and Response to Inflammatory Stimuli under Normal Healthy Conditions

KCs play a crucial role in protecting the body from external agents through mechanical and innate-immunity antimicrobial mechanisms. KCs participate in the regulation of the microbiome through lipid production and closely interact with the immune system. At the same time, their clasping interaction with sensory neurons and their role are discussed as important [17,18,19]. The tight interweaving of several systems at the body–environment border enables one to consider the skin as a unique neurological–immunological–cutaneous–endocrine organ [20]. Here, we review some of the mechanisms common to healthy skin that can play a role in inflammation and that are crucial for the development of the pathology of EBS.

First, KCs are under the influence of growth factors (GFs), which stimulate growth, proliferation, and migration, and can act in an autocrine (HB-EGF; TGF-β; amphiregulin; GM-CSF; CXCL-1, -8, -10, and -11; and CCL-14, -17, and -27 (C-C motif ligand 14/17/27)) and paracrine (EGF; TGF-β; neuregulin; IGF-1; FGF-1, -2, -7, and -1; HMGB; HSP90; GM-CSF; CXCL12; and C-C motif ligand 22 (CCL22)) manner (Figure 1) [21].

Second, crosstalk between the cutaneous peripheral nervous system and skin cells affects epidermal homeostasis [26]. The connection between sensory neurons and skin cells occurs via the participation of a variety of molecules (neuropeptides, neurohormones, and neurotrophins) with their specific receptors expressed by both neuronal and non-neuronal skin cells. Released neurotransmitters play the role of immunoregulators, exerting mitogenic influences on KCs. Neurotrophic factors and neuropeptides are also well-known as autocrine–paracrine factors for KCs. They have several important functions in the skin, apart from the induction of KC proliferation, which are the regulation of wound healing and the modulation of the immune response. In the wound-healing process, KCs are both effectors and targets of these factors [22]. In general, we can say that nerves and KCs are mutually stimulated. The most abundant skin neurotrophin is the nerve growth factor (NGF), which is mostly expressed in stem KCs. It stimulates KC proliferation and migration and participates in a positive feedback loop with neurotrophin-3, which also enhances KC proliferation. KCs can express receptors for neurotransmitters (e.g., adrenaline, noradrenaline, dopamine, histamine, and acetylcholine) [27]. Among neuropeptides, calcitonin-gene-related peptide (CGRP) and substance P (SP) are known to stimulate KC proliferation and induce KC inflammatory responses. Sensory neurons of the nociceptor that produce CGRP and SP are also involved in skin Th17 activation, which could contribute to the maintenance of chronic pain and inflammation [28]. This type of T-cell activation will be discussed later in connection with the pathology of EBS (see Section 9.3).

Third, the expression of a variety of Toll-like receptors (TLRs) is inherent to KCs. TLRs are expressed on multiple cell types populating the skin: KCs and Langerhans cells in the epidermis, resident and trafficking macrophages, T and B lymphocytes, endothelial cells, and skin microvascular cells. As important regulators of the host defense system, TLRs participate in the pathophysiology of inflammatory cascades. A role for TLR4 activation in KC injury has been shown [29]. KCs express TLRs 1–6 and 9. Ultimately, roles for TLRs 2, 4, 5, and 9, in synergy with IFNɣ, have been shown in different skin diseases associated with inflammation [30]. TLR signaling promotes the secretion of the cytokines IL-1, Il-6, TNFα, IL-8, CXC chemokine ligand 8 (CXCL8), C-C motif ligand 2 (CCL2), and C chemokine ligand 20 (CCL20). We discuss below (Section 9.2) the involvement of chemokines in EBS [31]. KCs express a vast number of cytokines and chemokines, which participate in paracrine and autocrine signaling (Figure 1). KCs secrete interferons (IFN), mainly IFNk and IFNβ; KCs pretreated with IFNγ induce the Th1 or Th17 differentiation of naïve T cells through CD2 activation. The treatment of KCs with mechanical stimulation activates the secretion of inflammatory cytokines (IL-1α, IL-6, IL-23, and TNF) and chemokines (CXCL1 and CCL20) [23]. When KCs express CCL2, they recruit macrophages for tissue remodeling. The engaged macrophages, in turn, activate KCs through EGF expression, which accelerates the locomotion of KCs.

One important trait of KCs is their dynamic nature, which is clearly demonstrated in wound-repair processes. Injury-triggered physiologic wound repair involves the transient epithelial–mesenchymal transition (EMT) of KCs, which enables them to acquire the enhanced migratory capacity associated with wound healing and tissue repair [32]. When a skin injury occurs, the wound healing (WH) process starts, which is a natural remedial process in epidermal tissue. It consists of four main stages: hemostasis, inflammation, proliferation, and maturation. In the acute stage of WH, lasting for the first few days after injury, the cytokines CLCX5, CLCX7, and CLCX8 are responsible for neutrophil recruitment [33]. The inflammation stage is characterized by the participation of peripheral blood mononuclear cells (PBMCs), resident skin cells, the extracellular matrix, cytokines, chemokines, growth factors, and regulatory molecules. During re-epithelialization, the KCs of the basal layer adapt, changing from a normal cuboidal shape into a flattened mesenchymal shape; they lose hemidesmosomes with a basal membrane and desmosome interconnections.

Chemokines are secreted from the sites of injury and play a key regulatory role in the WH process. In each stage of WH, chemokines are involved in the prevention (hemostasis) or promotion (inflammation) of angiogenesis [34]. The study of chemokine imbalance helps to elucidate the chemokines’ contribution to the diversity of healing processes specific to each disorder. CXCL4, which is released from α-granules in the epidermis, was found to be predominant in the early stages of normal WH; later, in the inflammatory stage, neutrophils are mainly recruited by CXCL8, CXCL1, and CXCL2 [35]. Macrophages replace neutrophils when the release of the chemokines CCL2, CCL3, and CCL5 recruits them to the wound. The involvement of macrophages is important, as they protect the wound from invading microorganisms and promote wound repair, secreting different growth factors, VEGF, bFGF, PDGF, TNF-α, and IFN-γ. The wound-healing process has been well studied in rodent models, but only limited studies have been conducted with EB lesions.

## 3. Epidemiology

The prevalence of epidermolysis bullosa simplex (EBS) is estimated to be approximately between 6 and 30 per million live births [36]. The statistics for all types of EB show a prevalence of approximately 1:50,000 births in the USA, among the patients registered in the Epidermolysis Bullosa Clinical Characterization and Outcomes Database (EBCCOD), 30.1% of whom have EBS, with signs of disease present at birth in 38% of these [37]. Precise and accurate estimates of the incidence and prevalence of EBS in the USA and Canada collected by the National Registry of EB (NEBR) showed a prevalence of 6.00 per million population and an incidence of 7.87 per million births. Approximately the same epidemiological estimates were found in Australia, while the incidence and prevalence of EBS were significantly higher in Scotland, Northern Ireland, and Norway and lower in Japan [38]. EBS was frequently diagnosed in the Netherlands (Dutch-EB-Reg data; 45.7%), with an incidence of 17.5 per million live births and a population-point prevalence of 11.9 per million. Dominant EBS cases were more common than recessive ones (90.1% vs. 9.9%) [39]. According to Orphanet, the prevalence of non-Dowling-Meara generalized EBS and localized EBS (ORPHA:79399) in Scotland is 1/35,000. The reported prevalence of severe EBS (former generalized severe, EBS gen.sev.) (EBS sev.) in Scotland is 1/1,700,000.

## 4. Brief Phenotypic and Genotypic Characteristics

### 4.1. The Basic Pathology of Most EBS Subtypes

The stability and resistance to mechanical stress of the skin are primarily determined by the composition and organization of the ECM in the dermis, but the epidermis makes a major contribution to the integrity of the skin, the epidermal keratins being the main proteins of the cytoplasm of KC. Keratins, as a mechanotransductive system that interacts with hemidesmosomes, are involved in the cellular response to ECM alterations [40]. The normal skin composition of the extracellular matrix (ECM) and homeostasis is different from that of injured skin and of skin with genodermatoses. Histological sections reveal epidermal cleavage and a loss of integrity, localized near the basal layer of KCs [41]. More than 70% of EBS patients have mutations in the genes *KRT5* and *KRT14* [38]. The respective products of these genes, keratin 5 (K5) and K14, are specifically expressed in these cells in the form of obligate heterodimers. EBS is a group of tissue fragility disorders with different forms that vary in severity. Blisters and erosions in the basal or suprabasal layer of the epidermis develop after minor friction or mechanical trauma. Generalized or palmoplantar blisters formed at birth or soon after birth and the development of keratoderma are found in all the EBS subtypes, with the cytolysis of basal KCs and intracellular keratin clumps being characteristic of more severe EBS forms (Figure 2).

The phenotypes of EBS sev. are characterized by clustered herpetiform blisters that arise spontaneously or as the result of minimal trauma, exhibiting palmoplantar keratoderma and nail dystrophy (Figure 2F) [42,43]. The development of basal cell carcinoma in the population with severe forms of EBS is slightly increased, with a cumulative risk of 43.6% by the age of 55 years, compared to the healthy population. Severe intraepidermal and mucosal fragility is accompanied by extracutaneous manifestations, which may cause disorders in other organs, e.g., muscular dystrophy and pyloric atresia in cases of EBS with plectin deficiency. These syndromic forms and other comorbidities, such as esophageal and intestinal epithelium malfunction, hinder the lives of patients with EBS despite the overall good prognosis and lifespans in cases of EBS. An important trait of EBS’ time course is the age-dependent decrease in severe disease manifestations. This could explain the compensatory effects of the expression of other nonmutant keratin and the partial involvement in the K5/K14 filaments of heterohybrid formation or the steady-state decrease in activity in the basal layer of the epidermis in adults.

### 4.2. List of Genes Involved in EBS and EBS Subtypes

The basal and suprabasal forms of EBS are classified according to the level of blister formation and cleavage. Mutations in seven different genes were found to be responsible for EBS (Appendix A). Apart from the most common causes of *KRT5* and *KRT14* mutations, which have already been mentioned, EBS could be induced by mutations in the other genes, such as *Dystonin* (*DST)*, *Plectin* (*PLEC)*, *Exophilin 5* (*EXPH5)*, *Kelch-like member 24 ( KLHL24)*, and *CD151 antigen (CD151)* [44]. A fairly wide spectrum of clinical severity has been reported, including forms with mild blistering on the limbs and forms with severe extracutaneous manifestations. The latest EB classification distinguishes the following subtypes of EBS: severe and intermediate subtypes, generalized or localized blistering forms with (frequent) autosomal dominance, and the subtype with (rare) autosomal recessive inheritance. Mutations in *KRT5, KRT14,* and *PLEC*, found in both cases with autosomal recessive and dominant forms, are discussed below. Mutations in genes (*KRT5, KRT14, PLEC,* and *KLHL24)* are known to be inherited in an autosomal dominant manner and induce the most common clinical subtypes of EBS. Generalized forms of EBS consist of the most severe subtype that was previously referred to as Dowling-Meara (EBS-DM) and the intermediate form of EBS with the former name Kobner (EBS-K). The localized form with the former name EBS-Weber-Cockayne has a milder manifestation. The several main forms of EBS are listed below according to the latest classification of inherited epidermolysis bullosa [44]. See Appendix A for information on the EBS subtypes.

### 4.3. EBS Subtypes

Common EBS subtypes:**Localized EBS** is characterized by skin blistering that develops any time between childhood and adulthood and is usually limited to the hands and feet. Later in life, the hand palms, and feet skin may thicken and harden (hyperkeratosis).**Intermediate EBS** is associated with widespread blistering that can be present from birth or develop in early infancy. The blistering tends to be more severe than that in localized EBS but milder than that in severe EBS.**Severe EBS** is the most severe type of EB simplex, in which extensive blistering can occur anywhere on the body, including the inside of the mouth cavity. Blistering tends to be present from birth and may improve with age, but older individuals can also be affected by hyperkeratosis. The severity and extent of the blistering vary greatly, and it can be fatal in infancy in very severe cases.**EBS with mottled pigmentation** is the fourth type of EB simplex, in which skin fragility is present at birth and, over time, brown pigmentation interspersed with spots develops on the body. The pigmentation can reduce and disappear in adult life.

Syndromic forms (with extracutaneous manifestations).
**Severe EBS with pyloric atresia (EBS-PA);****Muscular dystrophy EBS with PLEC deficiency (EBS-MD);****EBS with migratory circinate erythema (EBS-MCE);****Intermediate EBS with cardiomyopathy (EBS-CM);****EBS with nephrotic syndrome due to CD151 mutations.**

### 4.4. Mutations as the Causes of Pathology

Mutations in keratins lead to disturbances of the cell keratin intermediate filament (KIF) network and the development of tissue fragility, causing genodermatosis. K1 and K10 mutations induce the clumping of filaments in the suprabasal KCs in cases of bullous ichthyosiform erythroderma, while pachyonychia congenita is caused by mutations in K6a, K6b, K16, and K17 [45]. According to the Intermediate Filament database, K14 mutations are more prevalent in the genotypes of patients with severe EBS and intermediate forms, and K5 mutations are more enriched in localized forms (Appendix A) [46]. However, among the mutations that cause severe EBS, K5 is considered to induce the most devastating changes in epidermal homeostasis and beyond [8,47]. In general, most K5/K14 mutations are inherited in an autosomal dominant manner and induce the specific “dominant” features of the disorder; the targets for treatment include ameliorating these effects or eliminating the misfolded proteins [48,49]. The most dangerous mutations consist of the substitution of conserved amino acid residues, which weaken both the cell cytoskeleton itself and the cell–cell junctions [50]. The recessive mutations in *KRT5/14* have also been described; they are rare and also result in skin fragility. Patients with recessive *KRT14* mutations have a much higher incidence of the disease. K14 loss-of-function mutations lead to mild or severe forms of EBS [51]. Only a few cases caused by recessive *KRT5* mutations have been reported, and all the patients had either the severe forms or died neonatally [52]. A recent case report is worth mentioning, which examined a patient with two dominant heterozygous mutations, K14 I377T and K5 G138E. The patient had a localized form; however, severe signs were found on the soles [53]. Interestingly, K14 haploinsufficiency also causes two heritable diseases: Naegeli–Franceschetti–Jadassohn Syndrome and Dermatopathia pigmentosa reticularis, accompanied by the apoptotic death of basal KCs, hyperkeratosis, palmoplantar keratoderma, and other generalized manifestations. Mutations causing these syndromes are mainly localized in the N-terminal “head” domain [54,55,56].

Among the other genes involved in acquiring the EBS phenotype is the newly found *EXPH5*, which encodes an effector protein, Ras-related protein Rab-27B [57]. This protein is thought to be involved in exosome secretion and intracellular vesicle trafficking. Reduced expression of this gene results in keratin filament defects. Recently, another case of the abnormal regulation of ubiquitin-ligase Kelch-like family member 24 (KLHL24) was found, which causes EBS manifestations connected with K14 proteasome degradation [58,59]. The majority of EBS patients with these particular mutations also suffer from dilated cardiomyopathy (DCM), but the pathological mechanism of the disease in the heart is unknown [60].

Desmoplakin links the desmosomes to the keratin cytoskeleton, whereas plectin links keratin filaments with HDs. The type I HDs are composed of plectin, integrin α6β4, bullous pemphigoid antigen 1 isoform e (BPAG1e, also called BP230), bullous pemphigoid antigen 2 (BPAG2, also called BP180 or type XVII collagen), and the tetraspanin CD151. Mutations in any of these six genes induce EBS. Autosomal recessive forms of EBS are induced by mutations in *BPAG1-1e* and *EXPH5* genes [61,62].

Plectin is an important versatile cytolinker protein linking the cytokeratin network with hemidesmosomes (HDs); it is known to mediate the connection between junctional attachment complexes and intermediate filaments. A role for HDs, and particularly plectin, in reducing the cellular traction force and tension of KCs was demonstrated in a comprehensive model of EB-junctional KCs (PA-JEB) deficient and proficient, respectively, for β4-integrin (PA-JEB/integrin β4+) [63,64]. Moreover, plectin has the capacity to bind with β4-integrin and microtubules. Specifically, the inhibition of plectin and β4-integrin enhances focal adhesion (FA) formation, as well as the size of FAs, and cell spreading. Plectin is a multidomain protein with a wide expression profile in stratified squamous epithelia, muscles, and the brain. The uniqueness of plectin’s pattern of expression is determined by the multiple splice forms of the gene. For example, one of these forms, plectin 1b (P1b), has mitochondrial localization, while the other participates in the binding with nesprin-3. Recent findings of plectin directly binding with nesprin-3, a component of the outer nuclear membrane, provides clear evidence of plectin’s role in the regulation of the nuclear mechanics in KCs [65].

Mutations in *PLEC* may induce recessive and dominant forms of EBS. The diseases induced by mutations in the human plectin gene are autosomal recessive muscular dystrophy (EBS-MD) and pyloric atresia (EBS-PA). The specific isoform of plectin expressed in EBS-MD results in a form lacking the rod domain, which is encoded by exon 31 [66]. Pyloric atresia EBS (EBS-PA) is induced by mutations leading to the absence of the full-length and rodless forms of plectin. The dominantly inherited mutation in the rod domain of plectin leads to EBS-Ogna [67]. *PLEC* deletion induces aberrant cytokeratin organization and the activation of stress-activated protein kinases (SAPKs), c-src, and PKCdelta [68].

## 5. K5/K14 Heterohybrid Structure and Function

Keratins are classified into distinct types: type I Kty I (acidic) and type II Kty II (basic) are the major types. These are subdivided into types based on their sequence homology and biochemical properties. Keratins exist in cells as obligate heteropolymers consisting of polymers with one KtyI and one KtyII. The structure, which is universal for all keratin molecules, is formed by the conserved central alpha-helical rod domain [8,69]. The alpha-helical rod domain, with a helix initiation peptide (HIP) and helix termination peptide (HTP), contains four parts—1A, 1B, 2A, and 2B—with the linker non-helical parts, L1, L1-2, and L2, which interrupt a helical domain. Conserved heptad repeats (ABCDEFG) in the intertwined 2B coiled-coil domains form apolar interactions and salt bridges between residues, hydrogen bonds, and electrostatic interactions at the interface of the KtyI/KtyII heteroduplex. Two highly conserved motifs are found in IF: the first in the 1A domain and the second in the 2B domain 3′ part [70,71]. Coiled 1B subdomains of dimers overlap in the lateral interaction of dimers, forming tetramers, which are the key intermediates in IF assembly. The important part of the 2B domain TYRKLLEGE motif is a conserved C-terminal tail present in all IFs. It is considered to be involved in the assembly of coiled-coil dimers [70]. An additional feature of keratin’s secondary structure is the inter- and intramolecular SS bonds, which are necessary for polymerization and filament stabilization [70,72].

EBS disease-causing mutations can be divided into two major types, differentiated in terms of the keratin heteroduplex’s structure. These are interface mutations and surface-exposed mutations, with a stronger impact of residues, the side chains of which interact directly within the coiled-coil interface. K5/K14 mutations have been found located in eight distinct regions of molecules, with more abrupt disease manifestations arising from mutations located within the central rod domain. These mutations comprise long-range heptad repeats, organized in coiled-coil structures [8]. The reported crystal structure of the interacting 2B regions from these domains of the K5/K14 heterocomplex revealed asymmetric salt bridges, hydrogen bonds, and hydrophobic contacts in the interface of the K5/K14 complex, and its surface exhibited a notable charge polarization [70]. Most (more than 60% of) mutations are located in two places, in the regions of the helix initiation and helix termination peptides and in the region of trigger motives. Functional studies revealed that these are the areas directly involved in KtyI/KtyII interaction during the organization of the heterodimer structures [8,71,73,74]. Mutations in these areas are responsible for the most severe EBS variants.

The expression of the keratin 5 (K5) type I–keratin 14 (K14) type II “pair” is specific for the basal cells of squamous stratified epithelium. K5/K14 expression can also persist in suprabasal skin KCs before their entry into differentiation. Additional areas of K5/K14 expression are basal mucosal epithelial cells, including proximal regions of the digestive and respiratory tracts, individual cells in pancreatic and salivary gland ducts [75], and cells of the limbal corneal region and transitional epithelium of excretory and genital systems [76]. K5 expression has been described in individual medullary epithelial cells of the thymus [77]. In general, K5 is a marker of mitotically active and regenerative progenitor cells [78]. It has been found in the lungs, respiratory tract, urinary excretory system, prostate, and salivary glands [79,80]. This makes the K5/K14 complex an important participant in epithelial organ morphogenesis and regeneration. For the same reasons, K5/K14 are markers of oncotransformation and are used to detect lymph node micrometastases [81]. Upon skin damage, the activation of epidermal cells is accompanied by a switch of keratin expression from K1/K10 in the suprabasal layer to K6, K16, and K17. The HaCaT and AW13516 cell lines were used to study K14′s function in epithelial homeostasis [82]. The expression of the K14 protein was ablated by RNA interference. K14-deficient cells demonstrated a reduction in cell proliferation, decrease in phospho-Akt levels, activation of the Notch1 cascade, and increase in the levels of involucrin and K1, which are known to be markers of KC differentiation.

## 6. Turnover of Keratins in Healthy and EBS Cell Conditions

### 6.1. Dynamics of the Keratins

The KIF network is highly dynamic and sensitively motile. Keratin filaments begin to assemble in the periphery of the cell near the focal adhesion contact; then, oligomers extend them and are transported toward the nucleus. In the perinuclear region, oligomers dissociate and are transported to the periphery, being recirculated in networks. Relatively stable filaments appear near the nucleus and in hemidesmosomes. The slow transport toward the nucleus is actin-dependent, whereas the fast bidirectional transport is microtubule-dependent [83]. Keratins arise de novo near the sites of adhesion and nucleate near the cell periphery. PTMs affect the solubility of keratin filaments, their assembly, and their disassembly, which supports the turnover of polypeptides without destroying the structure of the entire cytokeratin network.

The properties and dynamics of keratin turnover were evaluated in knock-in mice, in which fluorescence-tagged keratin 8 (K8) was produced [84]. Homozygous K8-YFP mice were developed normally and became an in vivo model for KIF generation and biogenesis. The dynamics of KIF turnover in interphase cells and rearrangements in mitosis were studied. Fluorescent dot structures of keratin appeared first in close proximity to the plasma membrane in the blastomeres of the early embryos; then, keratin filaments arose. The turnover of keratin as measured by FRAP experiments indicated the prevalent intrinsic turnover of keratin-positive structures in a 15-min interval, with the recovery of fluorescence. Cell division was associated with persistent dotted fluorescence. Seemingly, the reversible formation of fluorescent keratin granules was detected in the juxtamembrane areas in the cultured cells, and this remained during mitosis and under stress conditions [85,86].

### 6.2. Posttranslational Modifications (PTMs) of Keratins

PTMs of keratins affect KIF solubility, which is significantly altered by disease-causing mutations in keratins, partly due to phosphorylation, sumoylation, and acetylation defects [87]. Phosphorylation-dependent KIF network organization was previously shown for cultured cells [88]. Briefly, the rapid and reversible phosphorylation pathway determines the plasticity of KIF and its respective adaptability to physiological conditions [89] (Figure 3). Multiple phosphorylation sites on keratin are considered to be part of the cellular defense system, which prevents the unwanted phosphorylation of apoptosis factors by stress-activated kinases [90].

Other posttranslational modifications (PTMs) of keratins include ubiquitination, sumoylation, and O-GlcNAcylation; the balance of these processes, resulting in KIF cycling, is able to maintain the tight regulation of keratins’ connections with HDs, cell-matrix adhesions, and desmosomal cell–cell adhesions [96]. In particular, a positive role for the site-specific O-glycosylation of K18 residues upon its expression in simple-type epithelia (serines 30, 31, and 49 of K18) has been shown, namely, for the activity of the prosurvival Akt and PKC kinases, which protect cells against apoptosis and promote their adaptation to stress and injury [97].

One conserved PTM of keratins is sumoylation, which regulates filament formation and solubility. In the case of oxidative and apoptotic stress conditions, filaments reorganize with hyperphosphorylation and induce an increase in keratin sumoylation, which reduces keratin solubility [98]. Another conserved PTM is the acetylation of lysines in the rod domain of keratins, with the deacetylase SIRT2 serving as a global regulator of KIF acetylation and the rearrangement of the cytoskeleton. Two putative acetylation sites (Lys178 and Lys185) in the 1A domain of K5 have been found, in particular, to be destroyed by hotspot mutations in EBS. Whether the role of the Lys-to-Glu substitution mutation or the absence of acetylation in K5 is the key cause of the disease is yet to be determined [99]. The PTM of keratins plays an important role in pathological pattern recognition, which specifically arises during KIF collapse and aggregation in EBS. Thus, direct changes in PTMs can ameliorate genotoxic cellular conditions. For example, a plant sesquiterpene lactone, parthenolide, which has anti-cancer and anti-inflammatory properties and modulates sirtuin function, was found to affect keratin organization, probably by reducing the elevated Lys acetylation of mutant keratins [100].

### 6.3. Phosphorylation-Dependent Aggregation in EBS Cells

The intense crosstalk between the KIF function state and phosphorylation is well-known [101]. The regulation of phosphorylation and hyperphosphorylation during mitosis and apoptosis was studied; the correlation between the cell stress response and the phosphorylated state of KIF was evaluated. Phosphorylated keratin was detectable in the patient’s cells without hypo-osmotic stress. Under hypo-osmotic stress, the level of phosphorylated keratin (assumed to exist in a non-filamentous form) increased in the cytoplasm of mutant cells [102]. The heterozygous state of the EBS mutation, in this case, means that the effect of stress-dependent phosphorylation develops despite the presence of a wild-type keratin. The activation of the c-Jun N-terminal kinase (JNK) axis was previously demonstrated for EBS cells [103]. In 2018, Mugdha Sawant et al. demonstrated a direct role for the phosphorylated state of a particular amino acid residue in the context of the K5/K14 pair of EBS [104]. The phosphorylation of Threonine 150 (T150) in the head domain of K5 in cells containing aggregated K14 R125C mutant was reported. The expression of phosphomimetic T150D K5 mutants results in the prevention of KIF network formation in K14R125C-containing cells, whereas the expression of the phospho-deficient mutant T150A K5 causes a reduced network without branching and reduced turnover.

## 7. Stress-Mediated Cellular Responses

### 7.1. Mechanical Properties of EBS Mutated Keratins

The mechanical fragility of the EBS epidermis correlates with the abnormality of the keratin network structure [105]. The pattern of aggregation in EBS KCs was found to be mutation-dependent, with a tendency to be enhanced in K14-mutant cells [8]. A model was suggested for stretch-induced stress analysis when the normal KCs and EBS patients’ cells were compared whilst being subjected to oscillating stretch [106]. This model showed the appearance of short rings and solid aggregates of keratins in stretch-exposed cells. The changes were observed in the zones of HDs and desmosomes. Desmosomes were more diffuse in sites of denser keratin, and HDs localized near the keratin aggregates. Desmoglein was shown to be resistant to stretch-induced stress, whereas the localization of plectin and desmoplakin was affected by the aggregation of keratins. Plectin and desmoplakin were directly associated with keratin rings, and BP180, plakoglobin, BP230, and plakophilin were also colocalized with keratin fragments. Additionally, the study of KEB7 cells showed that keratin aggregation inhibited PKB/Akt in response to β4 integrin activating antibodies (3E1), while the extracellularly regulated kinase (ERC 1/2) axes were kept active through EGF-R transactivation [107]. Another important issue concerns the nature of the key pathways that determine skin fragility. Disturbances in the structural function of the keratin network have not been unequivocally identified in EBS. Mutations causing EBS compromise cell stiffness, which was shown in a model of keratin type I and keratin type II (KtyI/II) knockout KCs with stable expression of keratin 14 (K14), K14R125P [93]. Investigations of the biophysical properties of the filamentous network in the healthy KCs, expressing K14 R125P, revealed that the EBS network was rather sparse but not mechanically defective. No correlation was observed between the strength of force applied and the change in the F-actin microtubules’ morphology. Interestingly, a decrease in KCs’ stretch-induced necrosis was observed upon the latrunculin A treatment of cells, which led to the disassembly of the F-actin network [102]. By contrast, a study on stretch resistance performed on KCs with the knockout of the KtyI or KtyII loci, transduced with the K14 R125P or K5 E477D mutant forms, respectively, showed impaired resistance to mechanical strain only in the case of the K14-mutant form, but not K5 [93]. The mechanotransduction in the keratin network was later investigated in detail in the HaCaT line. Increased substrate stiffness, modeling the matrix in scarring, led to a serine/threonine phosphatase-dependent change in the keratin network. Atomic force microscopy analysis revealed increased KC stiffness in accordance with an increased stiffer substrate with an F-actin stress fibril response. The expression of K14 R416P in HaCaT led to stiffness in fewer cells and a reduction in F-actin stress fiber in the stiffer substrate. Particularly interesting is the reduced expression of lamin A/C, as a nuclear mechanotransduction factor, in the nuclei of K14 R416P cells. Contradictory results for lamin A/C expression were observed in a plectin-deficient line [40].

The study of KtyI−/− mouse KCs expressing either K14 WT or K14 R131P revealed that mutant forms of K14 result in weakened traction forces, which were measured according to the parameters of cell-generated wrinkles. The traction force was found to be impaired based on the elucidated reduction in the focal contact area, reduction in the percentage of cells attached and spread over the collagen gel, decreased RhoA activity, and decreased RhoA presence in the cell periphery. Notably, treatment with a RhoA activator, CN03, improved the diminished properties of K14 R131P. Moreover, FAK phosphorylation and peripheral localization were found to recede in the mutant line. The manifestation of the decrease in adhesion in cells with keratin disturbances was the decrease in the directionality in the collective migration test with increased cell velocity [63].

### 7.2. ER Stress and the Ubiquitin–Proteasome System of Degradation

Although keratin’s network-based recycling is independent of protein translation [108], keratins participate in the ubiquitin–proteasome system (UPS) of degradation and sustain cellular turnover and metabolic balance (Figure 3). The ubiquitin ligase substrate adaptors KLHL16 and KLHL24 mediate a keratin’s proteasomal degradation, including that of mutant forms, since mutations of these adaptors lead to the EBS phenotype [109]. Ubiquitinated and hyperphosphorylated keratins can accumulate if proteasomes are inhibited or overloaded [92]. A model system with K8 expression showed that the hyperphosphorylation of keratins in vivo can protect keratins from ubiquitination but not entirely prevent it [101]. EBS-specific K14R125C aggregates inhibit proteasomal function [7] (Figure 3).

EBS mutations of keratins were shown to alter the dynamics of keratin turnover and assembly (Figure 3). Mutant keratin proteins form an increased amount of aggregates. To uncover the mechanisms of EBS pathology, it is common to use patient-specific cell lines that display clinical and ultrastructural features characteristic of EBS-sev. A classic example is the KEB-7 model with a K14 R125P EBS mutation under stimulation with osmotic stress (Figure 4) [102].

EBS-specific aggregation patterns could be induced in model systems of healthy immortalized KCs either by transfection with an EBS mutant encoding dominant-negative forms of K5 or K14 or using the CRISPR/Cas9 genome editing system (Figure 5) [110,111].

The live-cell imaging of transfected cells with a plasmid encoding the fluorescent protein fused with mutant keratin (K14R125C) revealed the strong accumulation of keratin aggregates in the cell periphery. Interestingly, the keratin granules were found to be in dynamic equilibrium with soluble subunits, with a half-life <15 min, whereas the keratin filaments were extremely static. The phosphorylated SAPK (p38) was found to become colocalized with keratin granules in response to various stress situations, which was demonstrated by colocalized expression [88,95]. The aggregation patterns of the keratins in the cells were influenced by the activity of specific SAPK inhibitors or by the presence of the non-receptor mutant phosphatase PTP1B. This mutant was engineered for trapping the phosphorylated substrates [90,95,101]. Time-lapse microscopy observations revealed that keratin granules were also formed in the cell periphery and continuously translocated toward the nucleus, and they could fuse and increase in size. Aggregates have a short lifetime (15–30 min) whereby they translocate to the nucleus and then disassemble quickly near the cell center [112,113].

### 7.3. ER Stress and UPR in Inflammatory Cascades in EBS 

Although disturbances in the keratin network reduce the mechanical properties of KCs, the activation of a stress response appears to induce the majority of homeostasis disruptions. The chaperone Bip/GRP78, a member of the Hsp70 family, translocates from the membranes in response to ER stress and induces the unfolded protein response (UPR). The involvement of a mild UPR during the normal differentiation of epidermal KCs was shown [114]. A high and active UPR characterizes several genodermatoses, connected with the impairment of epidermal differentiation: Darier’s disease (DD), keratosis linearis with ichthyosis congenita, and keratoderma. In these cases, UPR activation is followed by enormous ER stress from the abnormal protein folding, leading to disturbances in terminal differentiation. Some common mutant EBS forms of K14 (K14 R125C, K14 V270M, and K14 L384P) inhibit the 20S proteasome subunit through the ubiquitination of aggregated proteins, which leads to ER stress, as evidenced by the upregulation of cyclin D1, c-Jun stress-kinase, and Bip and increases in caspases 3 and 8, leading to apoptosis according to annexin V [7]. On the other hand, the analysis of the K14 P125 mutation revealed that chaperones such as CHIP (carboxyl terminus of Hsc/p70 interacting protein or STUB1) Hsc/p70, Hsp70, and Hsp90 mediate aggregated-form ubiquitination and consequent proteasomal degradation [115]. Caspase 8 was reported to be increased in both the lesions and intact biopsies of EBS patients’ skin. This points to caspase 8 as the ultimate cause of KCs’ apoptosis and cytolysis in the EBS [116]. In spite of the fully penetrating traits of mutations in the KRT5 and KRT14 genes, the considerable intra- and interfamilial variability in EBS suggests a role for such factors as molecular chaperones and the ubiquitin–proteasome system in the modification of disease severity [117].

The abnormal upregulation of the UPR system presumably occurs in EBS cells. Relevant evidence for this comes from experiments on the effects of chemical chaperones on these cells, e.g., N-oxide dihydrate (TMAO) [48,95]. Attempts to implement another chemical chaperone drug, 4-phenyl-butyrate (4-PBA), approved for urea-cycle-disorder therapy, were not encouraging. 4-PBA treatment decreases keratin aggregation and IL-1β expression. However, it also decreases the expression of cell-contact proteins and induces tumor necrosis factor (TNF)-α expression and the NF-κB and Wnt pathways [49]. Data from transcriptomic research on EBS cells confirmed the involvement of the UPR in severe EBS, as the ubiquitin-conjugating enzyme E2K (UBE2K) and a variant of the stress-induced chaperone HSP90, HSP90AB1, were reliably identified to be more abundant in EBS (KEB7) than in healthy control (NEB1) KC lines.

## 8. Molecular Pathways Orchestrated in the EBS-Specific Profile

One of the leading roles in EBS tissue damage is played by TNF-α. This pivotal cytokine plays a key role in the maintenance of KC activation by inducing the production of a plethora of signaling molecules, from epidermal growth factor receptor (EGFR) to IL-1 receptor antagonist [7,87,118]. TNF-α acts in the epidermis upon injury through different pathways. First, it initiates caspase-mediated apoptosis, via a type 1 TNF-α receptor, TNFR1. The second pathway consists of ceramide production, stimulation, and the activation of arachidonic acid synthesis, which conveys a signal promoting MAPK (mitogen-activated protein kinase) cascade activation. The last pathway, triggered by TNF-α, is the activation of the TNF-α receptor-associated death domain (TRADD) and TRAF2, with the consequent activation of the transcription factors NF-kB (nuclear factor kappa-light-chain-enhancer of activated B cells) and C/EBPβ. NF-kB was found in the suprabasal levels of the normal epidermis, suggesting a role for it in epidermal differentiation. The expression of TRAF1 and TRAF2 is increased in differentiating KCs [119]. TRAF2 also activates the MKK1 and JNK pathways. Among the different targets of C/EBPβ is the K6 promoter region, the activation of which is a TNF-α-dependent cytoskeletal response to an inflammatory stimulus [118]. Apart from the TNF-α cascade, there exists a separate role for the transforming growth factor (TGF)-α-dependent regulation of the KC state. TGFα mediates EGFR dimerization, activates intracellular protein kinase interactions with other factors, stimulates Ras, and in turn, activates a cascade of three protein kinases, Raf1, MAPK/ERK kinase (MEK), and ERK1/2 (extracellular signal-regulated protein kinase 1/2) [87].

TNF-α-mediated apoptosis was shown to be the key pathway in HaCaT cells transiently transfected with the frequent EBS mutation K14 R125C, common in severe EBS [7]. K18, K8, and K14 sequester TRADD in epithelial cells, thereby determining the diminished capacity of the cells for TNF-α induced apoptosis [7,120]. This highlights the hypothesis that misfolded keratin has an impaired affinity for TRADD, facilitating the TNF-α cascade, which activates caspase-dependent apoptosis and cell death.

Investigations elucidated discrepancies in the roles of K5 and K14 in TNF-α activation (Figure 6). The haploinsufficiency of KRT14 due to a nonsense or frameshift mutation in the E1/V1 region of the gene resulted in the increased susceptibility of KCs to TNF-α-induced apoptosis. In this case, keratin deficiency mediated the proapoptotic activation of TNF-α and participated in the pathogenesis of a rare ectodermal dysplasia called Naegeli–Franceschetti–Jadassohn syndrome [121]. Meanwhile, the TNF-α cascade remained inactivated in the case of K5(−/−) KCs and knockout mice. This investigation also did not find significant differences in TNF-α levels in HaCaT and MCF-7 lines stably expressing K14 R125C [5]. Another study found that newborn mouse KCs expressing helical K14 domains fused with K10 head and tail domains were less affected by TNF-α-induced apoptosis [122]. However, a further factor of TNF-α induction due to K14 irregularities is the association with plectin. Plectin is the earliest substrate of caspase 8, whose translocation from mitochondria into the cytoplasm initiates the important step of death by receptor-mediated apoptosis. The cleavage of plectin by caspase 8 is followed by poly(ADP) polymerase, gelsolin, and keratin cleavages. The impairment of apoptosis induction could occur in plectin-deficient cells, and this idea is supported by the observation of this abnormality in plectin-deficient mice [123]. The skin of newborn mice with plectin deficiency showed no signs of apoptotic cell death according to TUNEL analysis [124]. The evidence for the close connection and direct binding of K14 to plectin supports the data on the abnormal regulation of this type of apoptotic death in K14 mutants. The negative regulation of TRADD by the IFN-γ signaling pathway was demonstrated in the model of the mouse macrophage cell line RAW264.7. IFN-γ stimulated the phosphorylation of STAT1-α, its nuclear localization, and the binding of STAT1-α with TRADD in the nucleus. An important aspect of IFN-γ’s regulatory actions is the attenuation of TNF-α-mediated apoptosis, which is mediated by stimulating the cytoplasmic/nuclear translocation of TRADD and regulating the availability of TNF-α.

The early experiments with mechanical stretching showed that human melanocytes and KCs reacted to this stimulus by the activation of both ERK1/2 and SAPK/JNK cascades [133]. A series of regulatory molecular events were observed after applying the mechanical stretching of cells on the deformable substrate. First, an increase in the phosphorylation rates for the components of the ERK cascade was observed. Second, mechanical stretching induced KC proliferation, as indicated by BrdU incorporation. Third, K6 expression was induced, which is specific for hyperproliferative and inflammatory conditions in the skin [134]. Early work on hypo-osmotic shock, which induced cytoskeletal changes in EBS KCs, revealed the activation of SAPK/JNK pathways [103]. The known downstream targets of the JNK pathway are ATF-2 and c-Jun, whose activation by phosphorylation was observed in EBS KCs upon hypo-osmotic stress with the same time course as the phosphorylation of JNK1 and JNK2. Moreover, the JNK pathway was constitutively activated (before stress) in cells harboring more severe mutations and also displayed more rapid activation kinetics than in nonmutants or cells with a mild form of EBS. The activation of c-Jun and ATF-2 triggers the promoter elements of activating proteins 1 and 2 (AP-1 and AP-2), promotes the formation of complexes with leucine zipper domains, and induces the transcriptional activity of many genes, particularly through positive-feedback JNK signaling. EBS patient KCs were shown to be protected from stretch-induced apoptosis due to the activation of ERK cascade phosphorylation [129] (Figure 6). It was shown that the activation of Akt rather than that of ERK was capable of promoting cell survival through anti-apoptotic effects [135]. The activation of PI3/Akt by insulin-like growth factor 1 (IGF-1) induces cell survival through the phosphorylation of BAD and blocking the TNF-α-dependent translocation of proapoptotic Bax-α from the cytosol to the mitochondria, which was previously shown in a model of cytokine-mediated injury to oligodendrocyte progenitor cells (OPCs) and lens epithelial cells [136]. The study of immortalized KCs from patients with K14 R125P and K5 E475G mutations showed a decreased sensitivity to TNF-α and TRAIL-induced apoptosis as well as to ionizing radiation (IR)-induced DNA damage (Figure 6). EBS KCs demonstrated persistently upregulated pH2AX and pBRCA1, markers of DNA damage and apoptosis, while ATM/Chk2 DNA damage signaling was downregulated after IR. The anti-apoptotic factors Bcl-2 and FLIP were also upregulated in EBS KCs. EBS KCs are adapted to chronic stress and proapoptotic activation, which causes changes that may increase resistance to external influences and increase the propensity for oncotransformation [137]. This hypothesis was supported with the example of a patient with generalized EBS due to K5 E477D; the absence of BCL2 expression in biopsy samples from invasive carcinoma areas was demonstrated in comparison with its presence in intact skin areas [138]. This mechanism of adaptation to TNF-α-induced changes could explain the difference in results between the transient and constitutive expression of mutant forms of K14 in cell lines.

EBS KCs are characterized by the activation of dual-specificity phosphatases, which play an important role in MAPK-cascade inactivation [139]. The cells of specific lines from two patients were subjected to hypo-osmotic stress and subsequent microarray analysis in order to investigate the differential expression patterns. Interestingly, the cell line from a patient with a clinically severe EBS, K14 R125P, displayed the fewest transcript differences, which accounted for the pre-activation state of stress before the hypo-osmotic shock. Among the interesting genes with differential expression for both EBS-specific cell lines were the cyclin-dependent kinase inhibitor 1C (CDKN1C) and Kruppel-like factor 4 (KLF4) proteins, which are specific for the maintenance of a non-proliferative state and for the differentiation of the KCs. The activation of IL-6, which regulates growth arrest, and phosphatases of the dual-specificity DUSP family (MAP kinase phosphatase, MKP) was observed in response to hypo-osmotic stress. The downregulation of MKPs in both EBS mutants occurred in the resting background state. 

Positive-feedback regulation occurs in EBS-specific cells in the case of the enhanced expression of mutant Krt14, which is mediated by AP-1. JNK phosphorylation was active in NEB1 with the extrachromosomal expression of wildtype K14, where the presence of the JNK 1, 2, and 3 forms was detected [140]. The imbalanced superexpression of K14 leads to an elevated MAPK cascade (Figure 6), which, in turn, augments the promoter activity of K14. Thus, the primary cause of severe EBS’ pathophysiology implies an imbalance in K14 expression, which induces the enhancement of JNK signaling through a feedback loop and brings KCs into proinflammatory states with hypersensitivity to stress. Data from a murine model expressing mutant K14 demonstrated that a decrease in mutant gene expression resulted in normal morphology and function of the epidermis. Thus, a plausible therapeutic approach could be the partial suppression of mutant allele expression, which may be more effective than the overexpression of the wild-type allele [4].

However, keratin aggregates are not the only reason for the pathological inflammation in severe EBS. The shift in the keratin expression profile, which develops with a positive-feedback loop, concerns more than the K14 expression. A case of severe EBS carrying the K5 E477D mutation, in which blistering with severe keratoderma and massive verrucous carcinoma developed, was reported [138]. No detectable keratin aggregates were observed in this patient’s skin. Increased expression of K6, 16, and 17, triggering KC inflammation, was combined with reduced expression of p120 (catenin delta-1) (Figure 6). The P120 regulation of expression was associated with destabilized HDs and adherens junctions in KCs, while linkage with the proinflammatory NF-kB cascade was previously described for epidermal immune homeostasis [128]. Among the most upregulated transcripts were KRT14 and KRT16, but KRT6b and KRT17 were also found to be upregulated in an investigation. The aberrant regulation of signal transduction and affected cellular pathways of severe EBS KCs (with the K14 R125P mutation) were studied by the subtractive hybridization of cDNA [141]. The promoters of keratins might be activated by the SA-1/2 complex of transcription factors, which are influenced by permanently active SAPK/JNK axes. The activated wound-healing state of EBS KCs is determined by the upregulation of both K6 and K16. The different KtyI/KtyII heterodimeric combinations, such as K5 and K6, and K14, K16, and K17 were hypothesized to explain the compensatory effect for the impaired resistance to mechanical trauma resulting from the EBS mutations, particularly in K14 R125P. The markers of the KC activation state are K6 and K16; their expression is not found in steady-state epidermal KCs [118]. The activated KCs begin to secrete TNF-α and other growth factors such as TGF-α. Infiltrating into wound sites, lymphocytes produce interferon-gamma (IFN-ɣ), which acts on KCs through autocrine and paracrine signaling, stimulating them to further secrete growth factors. At this stage, the KCs are stimulated to overexpress K17, which is also a prognostic trait of epidermal inflammation. The expression of K17 in the basal layer of the epidermis is indispensable for numerous processes, which are associated with the proliferation and differentiation of epidermal appendages, most particularly regulating the hair follicle cycle [142]. The specific pattern of phosphorylation activated upon injury makes K17, as well as K14, interact with 14-3-3σ molecules and participate in KC growth [143,144]. The positive regulation of K17 promoter activity in KCs is mediated by the transcription factors AP1 and SP2, glioma-associated oncogene homolog (Gli1/2), Nrf2, and numerous cytokines, such as Il-17, Il-22, and TGFα. The overexpression of K17 mediated by these cytokines was demonstrated for inflammatory conditions (in the case of psoriasis) and carcinogenesis (cervical oral squamous cell carcinomas) [145,146].

However, specific to EBS, the upregulation of K17 expression, which is a critical player in psoriasis pathogenesis, was considered to be associated with the more frequent occurrence of BCC in severe EBS [141]. The involvement of K17, K16, and K6 in KC hyperproliferation is well known [146]. In addition, K17 has been described as an inducer of T cells, as well as of some cytokines, in an autocrine positive loop [145].

Other work with whole-transcriptome gene expression analysis, which was performed in the same model cell line (KEB7) with KRT14 R125P, uncovered some new specific targets affecting the EBS profile of gene expression, such as the significant downregulation of K19 [14]. Among other aspects, the positive regulation of BMP signaling and negative regulation of the canonical WNT receptor signaling pathways were shown. The significant downregulation of three genes, glypican 3 (GPC3), Msh homeobox 2 (MSX2), and integrin-linked kinase (ILK), belonging to the BMP cascade underlined the importance of TGF-β-family growth factors in the maintenance of epidermal homeostasis and remodeling. The upregulation of WNT5a is worth mentioning because abnormal WNT5a signaling facilitates invasion by different tumor types and its expression was found to be upregulated in cutaneous squamous cell carcinoma (SCC) and basal cell carcinoma (BCC) [147]. Cases of carcinoma development have been reported for EB patients, particularly BCC occurrence in EBS sev. forms. This and a previous transcriptomic study underline the unbalanced expression of genes involved in fatty acid metabolism and the retinoic acid pathway, such as the thioredoxin interacting protein (TXNIP); the role of each pathway in the differentiation of KCs is well known [14,141].

## 9. Inflammation Initiation and Progression

### 9.1. Sterile Inflammation (SI)

Injury-induced inflammation is called sterile inflammation (SI). It occurs via the secretion of proinflammatory chemokines and lymphokines, which are produced by macrophages and neutrophils attracted to the sites of injuries [148]. It is generally believed that sterile inflammation arising as a result of cell death or injury has the same mechanisms and acts through the same receptors as observed in the immune response against microbial infection [149]. This process starts with endogenous damage associated with molecular patterns (DAMPs), which may be represented by host-derived nonmicrobial stimuli, i.e., cellular substances released in the ECM during tissue injury. Noninfectious DAMPs, e.g., the mutant keratin epitopes arising in EBS, are recognized by pattern recognition receptors (PRR) on innate immune cells, initiating the proinflammatory cascade. Various DAMPs released during cell necrosis are instigators of inflammation [150]. A pivotal role in the regulation of innate immunity is played by Toll-like receptors (TLRs). TLRs mediate the link between the appearance of pathogen-associated molecular patterns, which are often associated with injury, and the transcriptional activation of the NF-kB pathway and MAPK activation (Figure 6). DAMPs from external infections, injury, or endogenous alarm molecules, which might be the alternative form of fibronectin (FN-EDA) in the ECM or the products of laminin degradation, are recognized by TLRs [151]. The propagation of sterile inflammation occurs in response to extracellular Hsp70 (eHsp70) acting via TLR2 and TLR4, which are expressed on KCs, among others [152]. The activation of Hsp70 and Hsp90 expression and colocalization with phosphorylated K5 have been demonstrated for EBS cells [95]. Other DAMPs such as S100 proteins and HMGB1 were also shown to be triggers of EBS proinflammatory pathways [10] (Figure 6). Therefore, we can regard EBS as a UPR-induced and, consequentially, ER-stress DAMP-associated disease and look for therapy targets for drugs aimed at reducing the DAMP response [49,112,150].

The blockade of TLR4 delayed NHEK migration prevented SAPK phosphorylation and decreased IL-1β production. The activation of the innate immune system via TLRs may induce systemic immune complications in epidermolysis bullosa, which was recently shown with serum amyloid A (SAA) secretion in human epidermal KCs, enhanced in response to TLR1/2 and TLR2/6 activation [153].

IL-1α was shown to be the crucial DAMP released by necrotic cells. IL-1α mediates neutrophil infiltration and induces CXCL1 secretion, which was demonstrated for mesothelial cells [154]. IL-1α is known to induce the expression of CXCL1 in epithelial cells and fibroblasts, which, in turn, activate the signaling cascade (NF-kB and MAPK induction). It was shown that IL-1R function on non-bone-marrow-derived cells was necessary for the development of an inflammatory response [155].

The secretion of the proinflammatory cytokines IL-1β and IL-1α by epidermal cells of different types is an important part of EBS activating cascades (Figure 6). Being expressed in the cytoplasm in precursor form, they begin to be secreted upon activation during injury/damage. The release of IL-1β initiates the KC activation cycle, and KCs maintain this state by the sustained production of proinflammatory cytokines. ILs stimulate the secretion of granulocyte-macrophage colony-stimulating factor (GM-CSF), TNF-α, TGF-α, and amphiregulin. IL-1β activates the specific IL-1β receptor on the basal KCs, the latter, in turn, acquiring active proliferation and migratory capacities [127]. The small-molecule drug diacerein, which was shown to inhibit IL-1β secretion, was applied in the EBS KC model system. A decrease in K14 aggregates was demonstrated and was accompanied by a decrease in ECM-degrading proteins. A similar effect was demonstrated for monoclonal antibodies (mAbs) against IL-1β [127].

Fibroblasts also migrate into the sites of injury and secrete TGF-β, which also stimulates the contraction of fibroblasts in an autocrine manner, activating the secretion of extracellular matrix proteins. It also induces KCs to temporarily reduce the rate of growth in order to restore their normal conditions and the expression of K5/K14 [118].

An important feature of EBS inflammation is healing without scarring. In the EBS microenvironment, when BM remains intact and the whole inflammatory cascade does not involve the decomposition of dermal ECM and activation of fibroblasts, there is no scarring [156]. Due to the impaired KIF peculiarities of EBS, associated with weaker HDs and aberrant adherence, the re-epithelialization stage is faster, thus mediating faster migration of EBS cells in comparison with control healthy KCs [63].

### 9.2. Chemokines

One cross-sectional study searched for a peculiar cytokine pattern in the skin of patients with different EBs [157]. The results obtained underline the systemic character of EB disorders, which is rather obscured by manifestations of cutaneous inflammation. Increased IL-1β serum levels were found in EBS cases, but lower or no difference in IL-2 and IL-6 was found in comparison with DEB, which highlights the different nature of the inflammation in the two subtypes of EB. A positive correlation between the disease severity and increase in IL-6 and IL-12 levels in the serum was noted. In another study, altered chemokine balances in the EB patients’ serum was found; CXCL12 and high-mobility group box 1 (HMGB1) were significantly elevated, while CCL21 (C-C motif ligand 21) was decreased [9] (Figure 6). It was recently found that HMGB forms a heterocomplex with CXCL12, enhancing the ability of CLCX12 to recruit monocytes [158]. Recent data showed the presence of chemokines in plasma and blister fluid (BF) collected from patients with different types of EB [125]. A high level of chemokines was detected in all the subtypes independent of age, gender, and blister location. A difference in the investigations was found, concerning the content of BF from early blisters. Significantly higher levels of CXCL5, Gro-family chemokines, CXCL8, and CCL2 (C-C motif ligand 2) were found in cases of early lesions of EBS than in advanced ones. This difference was not observed in the cases of other types of EB. The contents of CCL15, CCL17, and CCL20 (C-C motif ligand 5/17/20) were also consistently higher in early EBS than in other EB types (Figure 6). These chemokines orchestrate the migration of leukocytes to the sites of wounds. Moreover, an in vitro assay of the directional migration of the PBMCs of healthy donors towards chemotactic gradients was performed. The majority of migrating myeloid cells expressed CXCR2, CCR2, and CXCR1. The immunophenotyping of the infiltrated leukocytes showed CXCR2+ cells to be prevalent in EBS lesions.

### 9.3. Immune Cells

The role of macrophages, neutrophils, and dendritic cells, so-called professional phagocytes, as well as resident T cells, is important for the induction of a rapid response of innate immunity, stimulating WH, and preventing chronic infection in EBS skin wound healing [159].

ECM structure and remodeling are regulated by the immune cells of the skin, which are specifically recruited to the sites of inflammation. A variety of different immune cells that populate the skin promote tissue repair after injury. Those found in the epidermis include Langerhans cells, macrophages, dendritic epidermal T cells (γδ), and memory T cells (Figure 6). Tissue-resident memory T cells (TRM) and skin CD8 +T (TRM) cells become dispersed within the basal level of the epidermis after being engaged from the blood by the initial injury or infection. When the skin biopsy of one patient with EBS-MCE was investigated, the areas of the border of an erythematous lesion in the dermis and epidermal–dermal junction zone were abundant with CD4+ and CD8+ T lymphocytes, some of which expressed the CD69 activation marker. Apoptosis was found to take place in the KCs of the basal and suprabasal layers of the EBS-MCE epidermis [132]. CD4+ T helper cells, resident in the skin, secrete IFN-ɣ, supporting a quick response against a pathogenic microorganism. The infiltration of CD4+ lymphocytes, neutrophils, and eosinophils werewas found in skin biopsies of a group of severe EBS patients after a histological analysis. A high number of type 17 T-helper cells and specific cytokines (IL-17, IL-21, and IL-22) were found compared with those in a healthy donor’s skin. Apremilast was applied, which is useful for other immune-system-related inflammatory diseases. In this case, apremilast decreased blistering [131]. J.E. Mellerio proposed, in a short review devoted to potential therapeutic approaches to EBS, the development of more specific targeted therapies for EBS, with anti-IL-17 agents, which would be capable of targeting the Th17 response [160]. The severe defects of K5/K14 in severe EBS contribute to a potential defect in the T-cell education process in EB patients [159]. The thymic and skin tissues have a primary function in establishing immunity. In both cases, KCs are indispensable for promoting and activating lymphocyte responses [16]. In summary, the high level of cytokines observed in EBS serum could be explained by the systemic immune dysregulation and mainly pro-inflammatory state of EBS homeostasis.

Langerhans cells’ (LCs’) upregulation in response to DAMPs or upon skin injury results in their migration from the epidermis to the lymph nodes, where the priming of adaptive immunity occurs. The disruption of TGF-β expression in inflammation also stimulates LC migration from the epidermis [161]. LCs become antigen-presenting cells and activate both CD8+ cytotoxic T lymphocytes and CD4+ helper T lymphocytes. Interestingly, abandoned LCs were found in the epidermis of K5(−/−) mice; LCs may have immunoregulatory and tolerogenic functions [11].

B cells secrete a number of cytokines such as IL-6, GM-CSF, IFN-γ, and IL-4 at the site of inflammation. B cells continuously recirculate through the skin, being nonabundant in uninjured human and mouse skin. However, their functions within the epidermal environment of EBS are still unknown. One hypothesis suggests the exacerbation of skin inflammation by impaired B-regulatory-cell (B-reg) migration [162]. A subset of IL-10+ B1-regulatory cells (B1-regs) migrate into inflamed skin in an α4β1-integrin-dependent manner, thus mediating WH and producing pro-inflammatory cytokines, possibly interacting with skin T cells and regulating immunosuppression by the secretion of IL-10 [163]. However, there are still no reported data on B-reg cells in EBS skin; the preliminary assumption about their participation in the inhibition of the skin inflammation process could be based on experiments conducted for other inflammatory skin diseases (IMQ-induced psoriasiform and DNFB-mediated delayed contact hypersensitivity). Two mouse models with a B-cell-specific inducible deletion of α4β1-integrin showed reduced homing of IL-10+ B-regs into inflamed skin and showed a significant increase in cutaneous inflammation manifestations, proved by clinical and histopathological analyses.

Additionally, the transfer of IL-10-competent B cells into B-reg-deficient mice leads to a significant improvement in the psoriasiform skin phenotype. Thus, these data show a key role for skin-homing IL-10+ B-regs in the suppression of skin inflammation, supporting the notion that B-regs are critical players in the cutaneous environment during inflammatory skin diseases.

## 10. Disease Modeling

The main modeling systems for EBS are human-specific cell lines and animal models, mostly mouse models. Although mice represent an in vivo model of the disease, a number of differences between the murine and human epidermis increase the difficulty of using this approach. The mouse epidermis is anatomically much thinner than the human one (only newborn mice have a thicker epidermis), and it exhibits different mechanical properties. The stratum spinosum of mouse skin is less abundant and, morphologically, has fewer rete ridges at the dermal–epidermal junction; the shiny layer of the epidermis is absent. At the same time, mouse skin has the panniculus carnosus and many more hair follicles with sebaceous glands. These are not all the differences in the epidermal structures of these species. Re-epithelialization wound healing after skin injury is rarely investigated in models of small furry animals; their thin layers of epidermis make modeling extremely challenging. Humans and pigs are closer in this respect, and the comparisons are more representative [164]. Although this makes the clinical relevance of the mouse models elusive, this model system has played a major role in clarifying the role of the ablation of particular keratins or gain-of-function mutations in EBS.

The first notable aspect is the input of knockout mouse models of K5 and K14, i.e., K5(−/−) and K14(−/−). A K5(−/−) mouse model was developed in which extensive K14 aggregates similar to those in EBS patients were observed [6]. Notably, K5(−/−) mice, in contrast to K14(−/−), were more severely affected and died shortly after birth, displaying a lack of keratin filaments in the basal epidermis. Interestingly, K14(−/−) mice were reported to have generalized blistering of the skin resembling that of human skin with EBS gen. sev., but the K14(−/−) mice survived better than K5(−/−), reaching 3 months of age [165]. The K5/K14 KO mouse models clearly demonstrated different subsets of affected proteins involved in the K5(−/−) and K14(−/−) genotypes and, therefore, differences in the acquired phenotypes. One of the major differences between the two model systems is the absence of a keratin network in the K5(−/−) model and the presence of a weak residual one in K14(−/−). The possibility that endogenous K15 compensates for the loss of K14 and forms a heterohybrid with K5 may account for the functional difference [165]. Cases of EBS with PTC inducing mutations in KRT14, which induces NMD of the KRT14 transcript, displayed the presence of a weak K5/K15 network in the basal layer of the epidermis. In the case of basal KCs with the loss of K5, massive areas of cytolysis were observed, but the remaining expression of K14 in punctate distribution was found to be colocalized with plectin. The induction of K6 in mouse embryos that are K5(−/−) begins in the cytolytic areas of the skin at E18.5 and increases gradually up to birth. By contrast, K14(−/−) mice displayed no induction of K6 expression. The local induction of K6 underscores the data showing TNF-α release in response to local KC cytolysis. The expression patterns of K16 and K17 were found to be unchanged in these genotypes, with the exception of the similar punctate distribution of K17, as was shown for K14. Using the murine model, it was shown that K17 and K15 could bind to K5 and form a KtyI/KtyII heterohybrid in the absence of K14, with an increase in K17 expression [166]. The differences between murine knockout models and human cases in EBS manifestation could be seen, for example, in that the rare patients with K14 null mutations have EBS intermediate form, whereas mice with K14(−/−) exhibited EBS severe generalized form [165].

Preethi Vijayaraj et al. [167] investigated the constitutive deletion of the keratin type II cluster (KtyII −/−) in mice, which led to the discovery of the key functional elements in the knockout embryo. All the embryos were lethal by the E9.5 stage and had signs of growth retardation, but the embryonic intra- and extra-epithelia sustained apical polarity and had small mislocalized desmosomes. Interestingly, these mutants displayed intact localization for adherens junctions. The growth retardation in the mutant embryos could be partly accounted for by impaired mTORC regulation.

K5(−/−) mice were shown to have increased numbers of Langerhans cells in the skin [126], and this finding was in accordance with the chemokine upregulation found in EBS skin. The chemokines C-C motif ligand 2/19/20 (CCL2/19/20) are known to attract Langerhans cells to the skin.

The doxycycline treatment of K5(−/−) mice prolonged their lives for several hours after birth. The RNA-seq analyses of skin after treatment revealed the downregulation of pro-inflammatory factors; the decreased expression of IL-1-1β, IFN-activated genes, matrix metalloproteinase 13 (MMP-13), and serine protease 12; and the upregulation of serine protease inhibitor expression and various procollagens, which mark the process of post-inflammatory ECM remodeling [5]. These findings became the basis for the idea of the therapeutic application of tetracycline antibiotics in clinical trials on EBS patients.

Since mutant K5(−/−) mice die several hours after birth, the conditioning knockout models represent a step forward for understanding the mechanisms of gene network function. Apart from the formation of blisters and lesions due to epidermal detachment, the epidermis of K5-Cre KO displayed a normal stratification and correct expression patterns for differentiation markers. Interestingly, the proliferation and survival of KCs were unaffected. The TUNEL analysis of newborn plectin-deficiency mouse skin showed no signs of apoptotic cell death [124]. The occasional presence of apoptotic cells in the basal KC layer of blister roofs indicated that, although cytolysis due to KC disruption above the HDs predominated, some still-intact KCs underwent programmed cell death. Sensitivity to mechanical trauma, lesional epidermal barrier function, and KC fragility were observed in mice with conditioning knockout of plectin [124].

Recent studies on mouse knock-in models with K14 C373A, generated using the CRISPR/Cas9 system, demonstrated the role of a cysteine residue of K14 in disulfide bonding, which affects the organization and dynamics of keratin IFs in skin KCs. The epidermal KCs of knock-in mice possessed increased proliferation, faster transit times, and altered differentiation. It is worth noting that mice homozygous for the C373A mutation were viable and had no visible signs of surface fragility, except in the skin of the ears and the tail, which demonstrated increased thickness and barrier defects. Proteomic analysis identified 14-3-3 sigma as a K14-interacting protein, which subsequently enhanced YAP1 protein nuclear localization. Thus, an important role for the Cysteine 373 residue of K14 in regulating KC differentiation was uncovered by means of Hippo signaling regulation [144].

The inducible expression of a mutant keratin 14 protein in knock-in mice was also studied. This was the first attempt to generate a mouse model mimicking EBS with dominantly inherited mutations of K14. The inducible expression of mutant K14 was achieved by breeding with Cre-positive mice; biallelic pups, mtK14neo/CrePR1, were treated with the drug RU486 and developed visible blistering in areas prone to mechanical trauma [4].

Mice with a knockout of the plectin gene were shown to be resistant to CD95-driven apoptosis [168]. The study of different mutations of plectin and, more precisely, of two isoform-specific (P1d and P1b) knockout mouse lines revealed the uncoupling of IFs from mitochondria and disturbances in plectin isoform P1b expression, which causes severe mitochondrial dysfunctions [168].

A knock-in mouse model was developed for the study of the effects of the plectin dominant mutation, known to induce EBS-Ogna [169]. Decreased numbers of HDs in the skin of EBS-Ogna mice were demonstrated, and the altered pattern of plectin expression was shown only in KCs but not in the muscle cells. The proteolytical degradation of plectin associated with HD’s pro-isoform P1a was demonstrated.

These issues elucidate the drastic differences in skin-repair processes between humans and mice, which, again, makes the murine model of EBS mostly predictable. Despite the significant value of animal models in revealing the mechanisms of skin disease and platforms for new drug design, such systems are distinct from human skin in many physiological and morphological aspects.

Recently, the recessive type of EBS was reported in rhesus macaques, thus representing a novel nonhuman primate model [170]. Animals that were homozygous for the KRT5 insertion variant were stillborn and had widespread loss of the epidermis. Several alternative models of EB based on zebrafish and Drosophila were reported. Several EB-related genes are expressed in zebrafish skin (including KRT5 and KRT14), and a model with mutations in type XVII collagen demonstrates an EB phenotype [171,172]. The EBS models in Drosophila revealed human KRT5 and KRT14 expression and the development of a keratin network. The introduction of mutated KRT14 has led to semi-lethality, wing blisters, and perturbed cellular integrity [173].

In addition to animal models, cell lines are convenient and widespread model systems. Patient-specific and patient-derived immortalized cell lines have previously been established and indispensable for evaluating the molecular patterns associated with given gene aberrations, the aggregation pattern of the keratin network, and specific cellular traits such as migration, adherence, and flexibility in stress proneness [87]. Models based on EBS cell culture (mainly KCs) could be the best option since they consist of human cells, can take into account the heterogeneity of the disease, and are cost-effective compared to animal models. The main disadvantage of these models is the main cell–cell interactions in monoculture. However, the small number of cell types in these models will soon be improved with the introduction of more complex skin models in research practice [174]. At the same time, EBS patients are predisposed to abnormal pre-activation of the immune system, and the skin inflammation should be regarded in concert with the whole organism’s homeostasis. The main drawback of cell line modeling is that it cannot capture the distorted immune response in EBS, which necessitates using animal models despite the profound differences between murine and human skin.

## 11. Abnormal Cellular Structures and Function

Keratins are the structural component of the cytoskeleton, but aside from their integrated building role, keratins are indispensable participants in cellular metabolism [87]. Proteins of the armadillo and plakin families, e.g., plakoglobin, plakophilin, and desmoplakin, are associated with desmosomes and adherens junctions. The C-terminal domain of desmoplakin interacts with intermediate filaments, while the N-terminal one interacts with plakoglobin and plakophilin. These interactions affect the reorganization of KIF during migration, differentiation, and responses to stress [68,96]. Desmosomes play an important role in epidermal integrity and, particularly, in tumorigenesis, since their constituent proteins are capable of signaling (for example, Rho-signaling). The disruption of desmosomes and adherens junctions participates in the epithelial–mesenchymal transition (EMT). EMT is implicated in wound healing and in the stages of tumors becoming invasive cancers. The reverse phenomenon of mesenchymal–epithelial conversion was shown to occur in cases of squamous cell carcinoma treatment upon blocking the epidermal growth factor receptor and preventing the tyrosine phosphorylation of desmosomal components, plakoglobin, and desmoglein 2 [175]. The enhanced expression of desmoglein 2 and desmocollin 2, together with desmosomal cadherin, resulted in a transition of cells from a fibroblastic morphology to a phenotype with more epithelial traits [176]. Keratins stabilize plectin–β4-integrin interactions and specifically result in the surface localization of β4-integrin. KCs with keratin knockout demonstrated increased migration and invasion with a loss of patchy hemidesmosomal assemblies, where β4-integrin and plectin were no longer colocalized [94]. Mutations in the K5/K14 pair (most particularly, the K14 R125P mutation) result in abnormal plectin localization and enhanced β4-integrin turnover.

The early onset of blister formation and the severity of tissue fragility clearly show that plectin is more important in linking keratin filaments to hemidesmosomes than BPAG1, as BPAG1-knockout mice grow to adulthood and display only mild blistering [177].

An additional pathological manifestation in EBS cells is the aberrant localization of mitochondria. The mitochondria of KCs from EBS patients with the K14 mutation p.R125C are scattered in the cytoplasm, whereas the mitochondria of nonmutant KCs occupy the usual perinuclear space [178]. The recruitment of mitochondria to the cell periphery in EBS cells might be a response to stress or to the cell shift toward a high energy balance. A decreased oxygen consumption rate and reduced respiration and ATP production (nearly 3-fold relative to the rate of healthy cells) were revealed [178]. This important finding provides evidence of a close connection between the organization of the cytoskeletal network and the metabolic state of mitochondria. This link may include altered reactive oxygen species (ROS) signaling; mitochondrion-generated ROS have been shown to be versatile mediators of epidermal morphogenesis and differentiation [179]. Interestingly, the opposite correlation with the respiration rate was found for knockout mutants, namely, KCs with KtyI deletion. The impairment of the lipid composition of mitochondrial membranes was shown in KCs with deleted KtyI, with dark inclusions being observed in their mitochondria. These aberrations included the involvement of some proteins participating in the electron transport chain and increased oxygen consumption [180].

Another important function of keratins is their participation in vesicle and organelle trafficking, particularly in regulating the melanosome distribution in KCs. This function was clarified after the delineation of the impact of KRT5 mutations in such pathologies as Galli–Galli disease (GGD), Dowling–Degos disease (DDD), and some forms of EBS [87], as, for example, EBS with migratory circinate erythema (EBS-MCE) with c.1649delG of KRT5 and EBS with mottled pigmentation (EBS-MP) with the mutations p.Pro25Leu or p.Gly138Glu in K5 [181]. Families affected by the haploinsufficiency form of DDD with reticulate hyperpigmentation due to KRT5 mutations were investigated. A diffuse melanosome distribution, normally posing around nuclei, as well as desmosomes and HD deviations, was observed [182]. Further research on the phenomena shed light on the putative mechanism of melanosome–keratin interaction. It was revealed that the “head” domain of K5, not of K14, interacts with Hsc70. This interaction appears to provide the possibility for the keratin network to be involved in clathrin-dependent transport, as well as ensuring the filaments’ stabilization [183]. A recent investigation using xenograft mice models elucidated that hyperpigmented xenografts showed K5/FGF2 enhancement and increased K5 expression in all the spinous layers (which was additionally described for DDD patients), whereas K5 was downregulated in hypopigmented xenografts. Moreover, K5 hyperexpression in KCs enhanced melanosome transfer and interactions with cocultivated melanocytes [184].

## 12. Microbial Infections

Infection is one of the problems that induce complications in WH in EB lesions. Bacterial colonization in areas with lesions and wounds may reduce the healing rate, compromising the patient’s health [15]. The continuous blistering and intradermal liquid formation in EBS skin provide a “breeding ground” for multiple microbial and viral infections, which can be life-threatening [185]. The most common infections are caused by the opportunistic pathogens *Staphylococcus aureus* and *Pseudomonas aeruginosa*; moreover, different strains of *S. aureus* can colonize chronic wounds located in close proximity [186,187]. Several stages of contamination have been considered: the contamination of the wound surface by bacterial inoculation, the colonization of wounds when bacteria are present in greater numbers but not impairing healing, and infection or critical colonization when wounds are incapable of healing with more than 105 bacteria/gram of tissue [188]. With respect to immunity, a contradiction between the proinflammatory forces of the EBS microenvironment and the anti-inflammatory defense host system exacerbates the process of WH. Moreover, the innate immune system, acting through Toll-like receptor 5 (TLR5), recognizes the microbial antigen flagellin and mediates the development of inflammation-triggered carcinogenesis in the skin [189]. Antimicrobial peptides (AMPs) are part of the natural innate immune system of the organism; the most prominent for epithelial tissues is cathelicidin (human cationic antimicrobial protein 18, hCAP18, the transcription of which is activated by vitamin D3), which was reported to increase host defenses and accelerate wound closure in another type of EB called RDEB [190]. A series of short and overlapped peptides derived from K6A were found to exhibit antimicrobial and cytoprotective activities, keratin-antimicrobial peptides (KAMPs). Their central 10-residue glycine-rich hydrophobic regions are supposed to be a key element in their bactericidal activity [191]. K6A-derived KAMPs are processed as a result of K6A degradation through the UPS [192]. Data on the roles of different keratins, including K14, in microbe–host interactions and immune-system recruitment are accumulating [193].

The thymus epithelial cells provide a microenvironment for T cell development, allowing the survival and maturation of the thymocytes [194]. The pair of K8/K18 was expressed in cortical thymic structures, whereas K5/K14 expression specifically marked medullary thymic epithelial cells (mTECs) [195]. The main role of mTECs is to establish the central T cell self-tolerance: they perform the negative selection of conventional αβT cells and generate intrathymic Foxp3+ Treg cells [194]. The bipotent epithelial progenitor cells, important for the development of the thymus, co-express K5 and K8 encoding genes. Future studies may show whether, as in the epidermis, the keratin function in these cells is altered in EBS [16,196].

## 13. WH and Carcinogenesis in EBS

Recurrent skin blistering causes the formation of chronic skin wounds that are seen throughout life in patients with severe and generalized intermediate forms of EBS. The expression of MMP-8 is induced by nerve growth factors secreted from KCs and allows nerve fibers to penetrate into the ECM, which has been demonstrated for other dermatoses [197]. As a result, a local increase in nerve fiber density in the interstitial matrix may account for the development of abnormal itch and pruritus, which constitute frequent complaints of EBS patients (Figure 6). Periostin is upregulated in wound-healing processes and has also been reported to be highly abundant in epidermolysis bullosa with cutaneous squamous cell carcinoma (cSCC) and in the epidermis of patients with atopic dermatitis [198,199]. A recent study demonstrated that the JAK/STAT pathway is a key regulator of periostin secretion in KCs, which induces the involvement of α5β3 integrin receptors in sensory neurons, thus forming a direct link between skin inflammation and the spinal cord’s tactile and pain circuitries, in cases of skin pruritus. Another MAPK-activated signaling factor characterizing EBS, associated with pruritus and atopic dermatitis, is the DC-activating cytokine thymic stromal lymphopoietin (TSLP). An increase in TSLP was found in KtyI−/−, KtyI−/− with K14 R131P models, and R131P models, and correlated well with the EBS severity in patients [130] (Figure 6). It was hypothesized that enhanced TSLP was involved in the formation of the tumor hematopoietic microenvironment [200]. Additionally, an increase in TSLP has been demonstrated in several SCC types [201,202].

Damaging infections complicate the wound-healing process, and repeated rounds of inflammation create an increased tendency to develop tumors. The abnormal activation of the BMP/WNT axis was mentioned above. Indeed, the combined effects of bacteria, chronic inflammation, and wounding trigger cancer development in the skin, which was shown in a model consisting of transgenic InvEE mice (expressing MAP kinase kinase 1 (MAPKKK1) under the control of the involucrin promoter). It is known that NF-kB activation is essential in the inflammation associated with tumorigenesis [189]. The ablation of TNFR signaling (TNFR1,2−/−), myeloid differentiation primary response gene 88 (MyD88), and Toll-like receptor 5 (TLR5) reduced the number of tumors that developed upon wounding in mice [41]. As mentioned above, EBS KCs are less susceptible to apoptosis, which makes them more liable to oncotransformation [129,137]. K5 expression is associated with cell proliferation and tumor formation and was found to be a KtyII partner for K17 in tumors [203,204]. Generally speaking, the abnormal pattern of keratin expression produces an impaired program of KC differentiation, which, in turn, is closely connected with carcinogenesis [205]. Krt14 expression also serves as a marker of neoplasia [206]. K5 and K14 are markers of malignancy and have been investigated as participants in the propensity for tumor development [207]. The interaction between K5 and β-catenin, the main mediator of canonical Wnt signaling, which promotes the displacement of β-catenin from the membrane, was revealed in a breast cancer model in the case of progestin induction.

## 14. Conclusions

Disease-causing keratin mutations act to decrease the traction forces of the epidermal sheet, compromise adhesion to the acantholysis in the basal layer, and impair collective migration. Keratin, acting as a multiple substrate for phosphorylation under stress, thereby prevents the unwanted phosphorylation of apoptotic factors by stress-activated kinases. Aggregation-dependent SAPK/MAPK phosphorylation produces DAMPs that mediate stress-response signaling, providing the foundation for EBS’ pathology. The activation of TLRs and TNF-α-dependent cascades with the participation of NF-kB activity is the important link in mediating the signaling of EBS KCs and stimulating the immune response. IL-1β-predominant secretion results in the development of SI, while other specific chemokines initiate WH and attract immune cells to the site of injury. The inflammatory cascades in EBS include the component of SI in response to DAMPs, with a tendency to be chronic with numerous rounds of wound healing, exacerbated by microbial infections, with a decline in the adaptive immune response. The infiltration of CD4+ lymphocytes in rubbed intact EBS skin and the high levels of neutrophils and eosinophils in EBS skin biopsies provide evidence of chronic inflammation in these patients. The enhanced levels of Th17-specific cytokines in skin lesions and altered balance of chemokines indicate possible therapeutic perspectives for specifically decreasing the chronic inflammation in the pathogenesis of EBS.

## Figures and Tables

**Figure 1 ijms-22-12446-f001:**
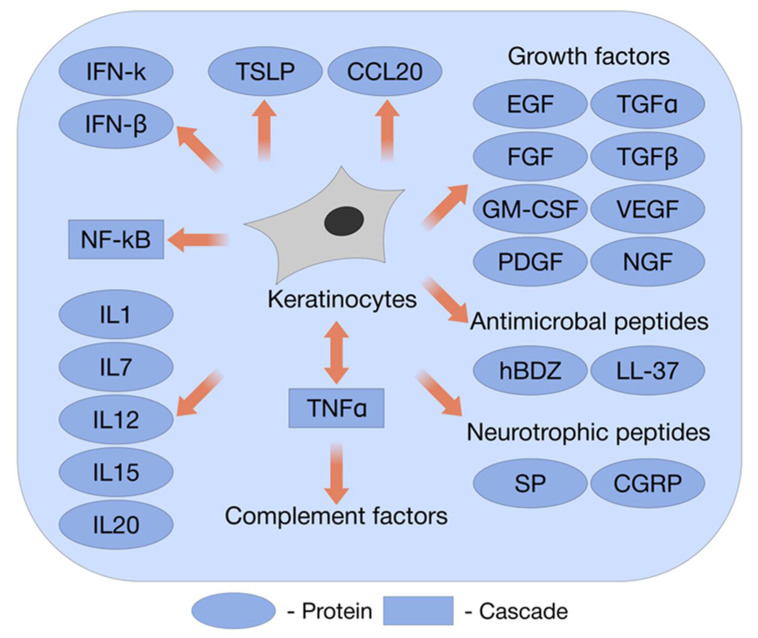
Scheme of key regulatory factors of healthy epidermal keratinocytes under normal and inflammatory conditions. Transforming growth factor-α (TGF-α), heparin-binding epidermal growth factor (HB-EGF), and insulin-like growth factor 1 (IGF-1) were shown to be essential for KC migration [21]. Different neurotrophins, with the most abundant being nerve growth factor (NGF), brain-derived neurotrophic factor (BDNF), and glial-cell-line-derived neurotrophic factor (GDNF), are involved in the regulation of KC proliferation, migration, and apoptosis [22]. The specific expression of the cytokines and chemokines C-C motif ligand 2, 20, and 22 (CCL2, 20, and 22) and CXCL1, 8, and 12 by KCs determines the immune response [23]. KCs manage the recruitment of the T-cell skin population to sites of damage through TNFα and IL-1α expression. They also realize a systemic shift in the T-cell response type through the thymic stromal lymphopoietin (TSLP) stimulation of Langerhans cells [24]. Neural calcitonin-gene-related peptide (CGRP) was shown to enhance keratinocyte proliferation and, together with substance P (SP), induce the inflammatory response of KCs [25]. Thus, the whole network of KCs involved in these processes is multifaceted.

**Figure 2 ijms-22-12446-f002:**
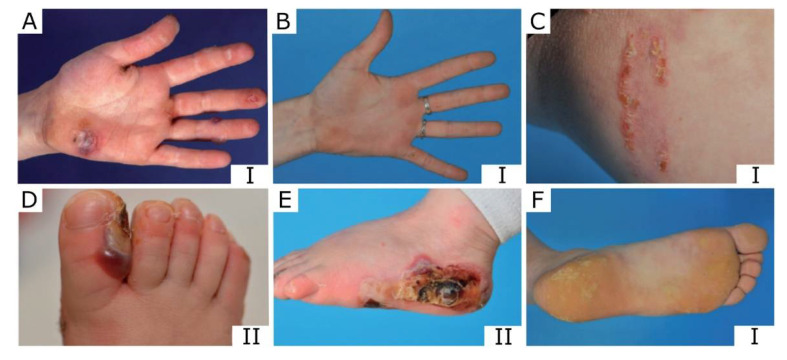
Clinical pictures of patients affected by epidermolysis bullosa simplex (I and II), generalized severe (EBS-sev.). (**A**) Blistering of the palm at 6 years of age with surrounding subtle hyperkeratosis. Bullae of the fingers with visible desquamation. (**B**) The same patient at 11 years of age, where amelioration of symptoms is seen. (**C**) Circinate blistering seen on the inner thigh. (**D**) Hemorrhagic blistering of the large toe, with focal hyperkeratosis and subtle onycholysis at 2 years of age. (**E**) Grouped hemorrhagic blisters on the lateral aspect of the foot of the same patient. (**F**) Generalized plantar keratoderma at 15 years of age. Images were adopted from [42].

**Figure 3 ijms-22-12446-f003:**
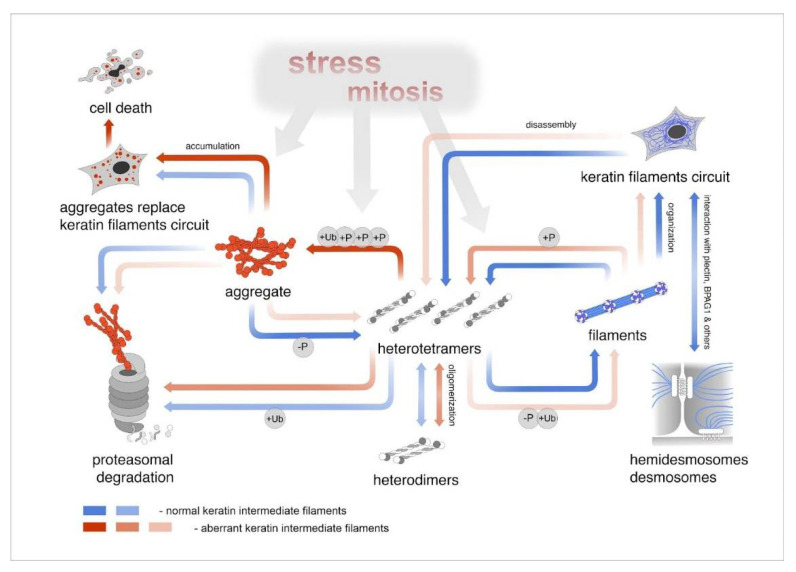
Keratin turnover and regulation of intracellular trafficking. Blue arrows indicate keratin circuit in healthy conditions; red arrows indicate the balance of mutant forms. The increase and decrease in color intensities reflect the relative strength of a process. Site-specific phosphorylation of keratins plays an important role in epidermal KCs [91]. The phosphorylated and sumoylated state of keratin determines its solubility and protects keratins from ubiquitination and proteasomal degradation [92]. As a result, a total decrease in cell adherence and hemidesmosomal/desmosomal interactions in the case of mutant keratins was demonstrated [93,94]. Aggregation of keratins is observed in EBS cells in severe types of the disease. The basal level of SAPKs (stress-activated protein kinases) is higher in cells derived from severe EBS cells than in healthy cells [95]. Aggregation of nonmutated keratin is connected with mitotic activity in healthy cells.

**Figure 4 ijms-22-12446-f004:**
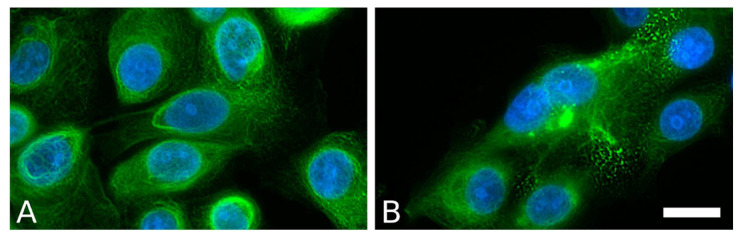
Effect of hypo-osmotic shock on the keratin cytoskeleton. (**A**). Normal immortalized cell line, NEB-1. (**B**). EBS-derived cell line, KEB-7. Both cell lines were subjected to hypo-osmotic shock and then fixed, permeabilized, and stained for K14 intermediate filaments. Clear peripheral aggregates were seen in KEB-7 cells 30 min after hypo-osmotic shock and remained for at least 3 h (seen in the picture). No filament fragmentation or aggregates were observed for NEB-1 cells. Scale bar = 22 µm. Images are adopted from [102].

**Figure 5 ijms-22-12446-f005:**
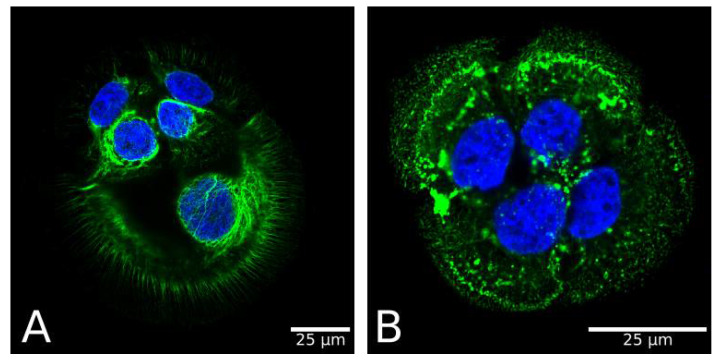
Wild-type and mutant HACAT D7 clone with disrupted KIF. (**A**). HACAT wild type. (**B**). Mutant HACAT D7. CRISPR/Cas9 technology was applied to modify KRT5 in HaCaT cells. Paired gRNAs specific to exon 7 of the KRT5 gene were used. WT—HACAT with wildtype K5; D7—mutation K5 L473P. Aggregates were observed in nondividing cells. Confocal images of immunofluorescence analysis of cells with antibodies against KRT5 (ab207351). Images are adopted from [110].

**Figure 6 ijms-22-12446-f006:**
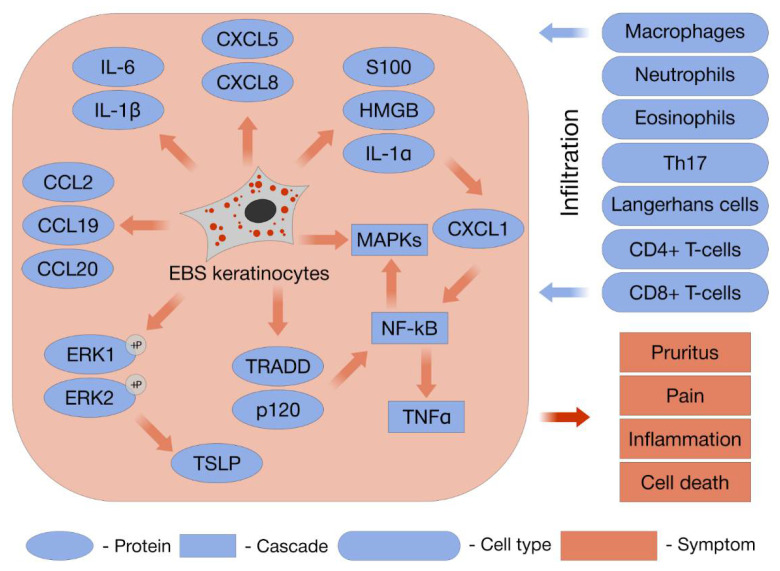
Scheme of key inflammatory participants in EBS skin. Aberrations in the keratin network of EBS keratinocytes activate stress signaling and trigger inflammatory and apoptotic signaling pathways, which lead to the secretion of pro-inflammatory agents. Acting through positive feedback mechanisms, proinflammatory agents contribute to the infiltration of different immune cells in the areas of EBS skin blistering [11,125]. This results in increased cell death, pruritus, and pain and induces chronic inflammation. IL-6/1β/1α—interleukins 6/1β/1α [126,127]; CXCL1/5/8—CXC chemokine ligand 1/5/8 [125]; S100—S100 proteins; HMGB—high-mobility group box [9]; MAPKs—mitogen-activated protein kinases [95]; NF-kB—nuclear factor kappa-light-chain-enhancer of activated B cells; TNF-a—tumor necrosis factor-alpha [7]; TRADD—tumor necrosis factor receptor type 1-associated DEATH domain protein [10,119]; p120—catenin delta-1 [128]; ERK1/2—extracellular signal-regulated protein kinase ½ [129]; TSLP—thymic stromal lymphopoietin [130]; CCL2/19/20—C-C motif ligand 2/19/20 [126]; Th17—T helper 17 cells [131]; CD4 +T-cells—T lymphocytes with CD4 receptors; CD8+T-cells—T lymphocytes with CD8 receptors [132]. Arrows indicate activation of an expression or process.

## Data Availability

Not applicable.

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
