# Peer review of "Keratins as an Inflammation Trigger Point in Epidermolysis Bullosa Simplex"

_ijms, 2021, doi:10.3390/ijms222212446_

Round 1
Reviewer 1 Report
Dear Authors,
Despite an actual topic touched in the review, I have some remarks and would like the authors to make some changes to improve the manuscript:
1) the short description of the normal keratinocytes, their factor description and inter-correlation with other factors in the epithelium/subepithelium is missed, - I would like to ask authors to develop such a subchapter and place it at the beginning of the review; some scheme representing keratinocytes factor expression/correlation with the proliferation, growth factors, ILs, neuropeptides might be involved here, too! Perhaps also the influence of skin neuro-immuno-endocrine system might be shortly mentioned here and the keratinocyte ability to change the phenotype, too!
2) move also the Epidemiology section forward after the Introduction and Normal skin description.
3) Conclusions. The first 2 sentences belong to the normal skin description and should be removed from here.
Thank you for the large References part. There are 7 previous century literature sources, but for such important review they just accept some historical issues, so, I do not insist to remove or exchange them.
Author Response
Dear Reviewer,
Thank you for your thoughtful analysis and recommendations.
Point 1: “the short description of the normal keratinocytes, their factor description and inter-correlation with other factors in the epithelium/subepithelium is missed, - I would like to ask authors to develop such a subchapter and place it at the beginning of the review; some scheme representing keratinocytes factor expression/correlation with the proliferation, growth factors, ILs, neuropeptides might be involved here, too! Perhaps also the influence of skin neuro-immuno-endocrine system might be shortly mentioned here and the keratinocyte ability to change the phenotype, too! ”
Response 1:
Indeed, our manuscript was lacking the description of normal keratinocytes. As you requested, we added the respective section at the initial part of the review as well as we added the corresponding scheme (Figure 1 in revised version). We describe shortly regulation of the growth and proliferation of epidermal keratinocytes. Response of keratinocytes to inflammatory stimuli in normal healthy conditions as well as the role of the skin neuro-immuno-endocrine system in these processes were also added. We are thankful for this valuable comment.
We marked all new sentences by yellow color.
Point 2:”move also the Epidemiology section forward after the Introduction and Normal skin description.”
Response 2: Right after this new chapter, we have placed the Epidemiology section with number 3.
Point 3: “Conclusions. The first 2 sentences belong to the normal skin description and should be removed from here.”
Response 3: We removed the first two sentences of the Conclusions since it was a piece of redundant information.
We hope that due to all these improvements you will decide that the manuscript deserves to be published in the «International Journal of Molecular Sciences».
.
Reviewer 2 Report
Sir,
I have recently reviewed the manuscript "Keratins as an inflammation trigger point in Epidermolysis Bullosa Simplex" submitted by Nadezhda A Evtushenko and co-workers to IJMS (ijms-1397834).
Surprisingly, during the first reading, I noted that the authors reproduced Fig.1 from a publication of Beriault, D.R. and-co-workers Plos one, 2012). They mentioned that it was "adopted" and included a reference to this work. But there is no explicit declaration that the authors of the original work and publisher granted their permission to such "cherry-picking". I believe that this is not tolerable behaviour in the scientific community.
Due to ethical concerns, this manuscript should be rejected.
Author Response
Dear Reviewer,
We would like to express our gratitude for your efforts in critical consideration of our review.
We understand your concern regarding the use of the figure from the article of Beriault, D.R.; Haddad, O.; McCuaig, J.V.; Robinson, Z.J.; Russell, D.; Lane, E.B.; Fudge, D.S. The Mechanical Behavior of Mutant K14-R125P Keratin Bundles and Networks in NEB-1 Keratinocytes. PLOS ONE 2012, 7, e31320, doi:10.1371/journal.pone.0031320. The authors have published this article under the terms of the Creative Commons Attribution 4.0 (CC BY 4.0) License. This License permits unrestricted use, distribution, and reproduction in any medium, provided by the original author and source. Since we have properly cited the original article in the main text and in the figure legends there is no ethical concern and the use of this figure fully correlates with IJMS rules for publications. Moreover, according to these rules, we could not include the original unpublished data in the review. We hope that you change your mind and recommend our manuscript for publication.
Reviewer 3 Report
The aim of this paper, entitled “Keratins as an inflammation trigger point in Epidermolysis 2 Bullosa Simplex” is to review the pathogenesis of this disease. The hypothesis of an inflammatory pathogenesis is intriguing, and several key ways are extensively discussed. However, few points need to be clarified before the publication.
The introduction could be implemented by a table that summarizes the characteristics of the different forms of epidermolysis bullosa. One or more clinical photos would also be explanatory. Moreover, a table could also clarify the mode of inheritance of the different EB forms (autosomal dominant, recessive, x-linked).
Some parts, such as the description of the structure of the epithelia, are redundant and should be reduced.
There are several typing errors, that should be corrected.
Author Response
Dear Reviewer,
Thank you for your thoughtful analysis and your valuable recommendations.
We try to make all the changes that you recommended concerning the addition of information and figures and we removed of redundancy sentences.
Point 1:” The introduction could be implemented by a table that summarizes the characteristics of the different forms of epidermolysis bullosa.”
Response 1: Thank you for this comment. Indeed, Epidermolysis bullosa is highly heterogeneous group of diseases, the mechanisms involved in pathogenesis and causative genes are quite variable among EB group. As our review is devoted to the problems of epidermolysis bullosa simplex, we tried to describe this very group leaving the other subtypes of EB out of frames of the manuscript. We are very grateful for your recommendation to describe the variety of disease subtypes in the table, which includes the targeted gene information and mode of inheritance. Therefore, as you suggested, we made a table that describes the different forms of epidermolysis bullosa simplex, and we put this new table in the Supplementary (Table S1). The reference on Table S1 placed in the Introduction and marked yellow.
Point 2: “One or more clinical photos would also be explanatory. “
Response 2. We fix the problem with the absence of figures with clinical features of EBS patients and put this figure at the beginning of the review We added a clinical picture that represents EBS injuries of EBS severe subtype at the beginning of the main text as well (Figure 2). Thank you for this comment, it improves the data.
Point 3: “Some parts, such as the description of the structure of the epithelia, are redundant and should be reduced.”
Response 3. We have tried to shorten all sections that might have contained redundant information.
Point 4:”Typing errors should be corrected.”
Response 4: We have performed a thorough spell-check and corrected all typing errors that we were able to identify.
We hope that we fixed most errors and met most of your requirements for publication in the «International Journal of Molecular Sciences».
Round 2
Reviewer 2 Report
Sir,
I have recently reviewed the second version of the manuscript "Keratins as an inflammation trigger point in Epidermolysis Bullosa Simplex" resubmitted by Nadezhda A Evtushenko and co-workers.
The authors provided their answer regarding the legality of the use of the particular figure in their manuscript. I acknowledge that it was probably absolutely rightful to use it- however, I am not scientifically convinced that this figure helps to improve the quality of the manuscript significantly. It is always advisable to present your own data (as proof of expertise). If the authors do not have it (e.g. due to the unique nature of the material), it seems necessary to discuss properly why it was necessary to take it from someone else. This is obviously insufficient in this manuscript.
The Introduction is poorly written. Some statements here are rather confusing (this might be just a language problem) or scientifically unsound. E.g., line 50 "The location of visible lesions in the epidermis makes EBS keratinocytes a suitable model for the investigation of disease mechanisms ..." The molecular pathogenesis of EBS was deciphered several decades ago and is broadly known. It remains obscure what do they mean by these words ...
Line 53: "adhesion processes in the cell-matrix". The meaning of this is unclear. Is it a stub?
Line 65-67: " Chronic inflammation and immune system pre-activation due to chemokines and cytokines are considered to be the prerequisites for the development of the EBS pathogenesis." NO. I am sorry, but this statement reveals that the authors do not distinguish cause and consequence in EBS. This failure (of the concept) is obvious throughout the whole manuscript.
Section 2 is a chaotic assembly of more or less relevant aspects of cutaneous biology, immunology, neurology... Moreover, some aspects remain controversial (namely keratinocytes having synapse-like structure with sensitive neurons"). This concept is not widely accepted and is opposed by many others. If this section should serve as a prequel for the "inflammatory aspects in EBS", I believe it would require far better data selection and more careful wording.
Section 4 - the heading is unsuitable. Actually, the content of this section would fit more under the heading of section 5. There is somewhat of cell pathology here.
Also, the caption of Figure 2 remains unclear to me. It is another "adopted" figure. The caption mentioned, " affected patients (III-3 and IV-1)". Is it relevant to the original article?
Section 5 - the selected heading is wrong. The text does not bring any data on cellular pathology. This section recapitulated possible target genes in EBS.
Subsection 5.2 represents only shortlisted rare forms of EBS. Without any description, without a single complete sentence!
Section 6 - why do the authors start this section called " Keratin expression profiles" with a brief notice regarding other secreted components? As seen in previous sections of this manuscript, the headline and the actual content are only loosely associated. BTW., in the whole section, there is very little said about keratin expression patterns.
Section 8 is (probably) the principal part of the manuscript. Unfortunately, this is mostly a recapitulation of earlier publications of other authors—very extensive, very exhaustive and not bringing much insight. The authors should here select only the most critical aspects and interpret these data in the context of their theory.
Section 9 - again, very extensive and not a concise assembly of basic immunological knowledge.
Section 10 - I am not sure why the authors placed this section here in this order? Is there any benefit of presenting these data now and here? Section 4 or 5 would probably accommodate this paragraph better.
I also believe that section 14 presents probably what was announced in the headline of section 7.
Finally, section 13 - this section summarises some data on inflammation and EB. However, I do not find here any emphasis on specific features in EB. Moreover, the interpretation of inflammation in this section differs from what was mentioned in the Introduction.
To conclude, the manuscript does not represent an important contribution to our knowledge of EBS at any rate. The manuscript is poorly structured. It has an obvious ambition to cover the topic extensively, but it fails in many aspects to do so. The authors should reconsider what should be the core of their communication. As it is, the manuscript does not have any clear narrative. It is lengthy and messy text. The extensive language revision by a native speaker (preferably with some background in biomedical sciences) would be beneficial. Also, I would recommend preparing some own figures. The cartoons provided by the authors have relatively minimal informative value and are of inferior art quality.
Author Response
Dear reviewer, thank you for so detailed presentation of your critical remarks. We tried to respond to each point.
We revised all the Sections of the manuscript according to your valuable comments and hope you consider the renewed version would be possible to publish in the Special Issue of the International Journal of Molecular Science.
Point 1
“The authors provided their answer regarding the legality of the use of the particular figure in their manuscript. I acknowledge that it was probably absolutely rightful to use it- however, I am not scientifically convinced that this figure helps to improve the quality of the manuscript significantly. It is always advisable to present your own data (as proof of expertise). If the authors do not have it (e.g. due to the unique nature of the material), it seems necessary to discuss properly why it was necessary to take it from someone else. This is obviously insufficient in this manuscript.”
Response 1
Keratin aggregation - one of the main features of EBS severe keratinocytes - and therefore the respective phenotype needed to be demonstrated.
We agree that it would be better to present our own data. That is why figure 5 of our manuscript represents our own data on keratin aggregation. Indeed, EBS is the orphan disease with low incidence, and it is difficult to obtain primary keratinocytes of EBS severe subtype patients, therefore at this point of time we had to use data from other scientific groups While discussing the aggregated KCs patterns in EBS we mentioned the well-known EBS severe mutation K14R125P and the respective patient specific cell line, KEB7, as the respective features of keratin’s network were published and are well-known to the scientific community. Moreover, according to our practice (we work with EBS- specific KCs) this clear aggregative pattern is not typical and not observed in KCs of other EBS forms. We included our data on artificial EBS model in HaCaT (Figure 5) in order to illustrate the aggregative capacity of mutant keratins and we used the classical example of KEB7 cell line for comparison and in order to underline the specific EBS sev traits.
Point 2
“The Introduction is poorly written. Some statements here are rather confusing (this might be just a language problem) or scientifically unsound. E.g., line 50 "The location of visible lesions in the epidermis makes EBS keratinocytes a suitable model for the investigation of disease mechanisms ..." The molecular pathogenesis of EBS was deciphered several decades ago and is broadly known. It remains obscure what do they mean by these words …”
Point 3
“Line 53: "adhesion processes in the cell-matrix". The meaning of this is unclear. Is it a stub?”
Point 4
“Line 65-67: " Chronic inflammation and immune system pre-activation due to chemokines and cytokines are considered to be the prerequisites for the development of the EBS pathogenesis." NO. I am sorry, but this statement reveals that the authors do not distinguish cause and consequence in EBS. This failure (of the concept) is obvious throughout the whole manuscript.”
Responce 2, 3 and 4
Indeed some sentences in the introduction were poorly written and difficult to grasp. We have made significant revisions on the mentioned and other parts of the introduction and hope that these changes will clarify our intentions more distinctly.
Point 5
“Section 2 is a chaotic assembly of more or less relevant aspects of cutaneous biology, immunology, neurology... Moreover, some aspects remain controversial (namely keratinocytes having synapse-like structure with sensitive neurons"). This concept is not widely accepted and is opposed by many others. If this section should serve as a prequel for the "inflammatory aspects in EBS", I believe it would require far better data selection and more careful wording.”
Response 5
In this section we discussed several aspects of normal skin physiology. This section was added according to the Reviewer's comment which concerned the importance of the neuro-immuno-endocrine system interactions. We appreciate your indication on the contradictory data on the formation of a synapsis-like structure in the sensory neuron dendrite and keratinocytes junction. Therefore we choose more precise expressions to describe it. Nevertheless, in our discussion of this system, we consider important to mention recent works in the field of neuro-epidermal interaction.
Point 6
“Section 4 - the heading is unsuitable. Actually, the content of this section would fit more under the heading of section 5. There is somewhat of cell pathology here.”
Response 6
Thank you for your comment. The section 4 and the heading have been redesigned. We rename it as “Brief phenotype and genotype characteristics” and reorganize it into 4 subsections.
Point 7
Also, the caption of Figure 2 remains unclear to me. It is another "adopted" figure. The caption mentioned, " affected patients (III-3 and IV-1)". Is it relevant to the original article?
Response 7
Yes, we have placed this figure here as an example of a clinical picture of EBS patients in order to help readers to understand the actual meaning behind clinical terms. “III-3” and “IV-1” were symbols for EBS patients and do not have other meaning behind that. Probably as it could be confusing for readers we changed them to “I” and “II”.
Point 8
Section 5 - the selected heading is wrong. The text does not bring any data on cellular pathology. This section recapitulated possible target genes in EBS.
Response 8
We decided to merge Section 5 and subsection 7.1. Thank you for your valuable critical comment! New Section 5 has the heading “K5/K14 heterohybrid structure and function”. In this chapter we presented the brief information on epidermal keratins structure, heteroduplex features, focusing mainly on the K5/K14 which is the main target proteins in EBS. We give here the information about expression of these keratins, merging some sentences from the former Section 10, “Functional role of the EBS-affected proteins”, but make this fragment shorter.
Point 9
Subsection 5.2 represents only shortlisted rare forms of EBS. Without any description, without a single complete sentence!
Response 9
In our second version of the Manuscript we included, indeed, the additional information concerning these forms, placing it in the Supplementary TableS1. Also we have mentioned rare forms of EBS associated with EXPH5, KLHL24 , CD151 in the former Section 7, subsection 7.1 ”Mutations as the causes of pathology”. Plectin associated EBS pathologies (EBS-MD and EBS-PA forms) are mentioned later in the former Section 7, subsection 7.1 ”Mutations as the causes of pathology”. In the present version, when we rearrange the structure of the manuscript, you can see this information in Section 4.
Point 10
Section 6 - why do the authors start this section called " Keratin expression profiles" with a brief notice regarding other secreted components? As seen in previous sections of this manuscript, the headline and the actual content are only loosely associated. BTW., in the whole section, there is very little said about keratin expression patterns.
Response 10
We thank you for this comment. Indeed, we decided to merge Section 6 and 10 in one chapter. The new Section 6 has the heading “Turnover of keratins in healthy and EBS cells conditions”, it contained subheading 6.1 “Dynamics of the keratins” and “6.2. “Post translational modification (PTMs) of keratins’” and 6.3 “ Phosphorylation dependent aggregation in EBS cells”. It contains data about expression of keratins.
We reorganized Section 7 - inserting new Section 7 with heading “Stress-mediated cellular responses”. It contained sections, concerning cellular stress and protein degradation: Subsection 7.1. Mechanical properties of EBS mutatated keratins”,7.2. ER Stress and UPR as inflammation cascades in EBS; and 7.3. ER Stress and UPR as inflammation cascades in EBS.
Point 11
Section 8 is (probably) the principal part of the manuscript. Unfortunately, this is mostly a recapitulation of earlier publications of other authors—very extensive, very exhaustive and not bringing much insight. The authors should here select only the most critical aspects and interpret these data in the context of their theory.
Point 12
Section 9 - again, very extensive and not a concise assembly of basic immunological knowledge.
Response 11 and 12
In these sections, we review many investigations concerning EBS, performed during a long period of time. The purpose of our review is to consolidate knowledge on EBS and to review the existing ideas concerning its pathogenesis. Despite the large number of papers in this theme, some aspects of these problems remain unclear because most works were aimed to study the similar mechanisms. For example, despite a large number of works on the cellular mechanisms of EBS, many of them were performed on lines with similar mutations. Therefore, we try to review the various works, which in some cases present controversial ideas. In addition, a small review of general immunological points allows us to draw attention to those issues in the inflammation processes, which remain unexplored in EBS pathology. In the new revised version of our paper the problem is displayed in Section 7 and in the new Section 8 entitled “Molecular pathways, orchestrated in the EBS specific profile”.
Point 13
Section 10 - I am not sure why the authors placed this section here in this order? Is there any benefit of presenting these data now and here? Section 4 or 5 would probably accommodate this paragraph better.
Response 13
Thank you for this comment! Your comment helped us to bring more logic in the manuscript. Indeed, we reorganized it completely, and decided to merge in Section 5 several pieces of the information from former Sections 6 and 10.
Point 14
I also believe that section 14 presents probably what was announced in the headline of section 7.
Response 14
We are grateful for your suggestion. Therefore, we moved this Section 14 to the Section 9, where pro-inflammatory cascades are discussed. Thank you for advice to rearrange this subdivision.
Point 15
Finally, section 13 - this section summarises some data on inflammation and EB. However, I do not find here any emphasis on specific features in EB. Moreover, the interpretation of inflammation in this section differs from what was mentioned in the Introduction.
Response 15
Thank you again! Section 13 is now Section 9. We shortened it a lot, leaving the most significant parts concerning the immune systems in EB and subdivided them into 3 chapters, which seems to be logically valid. We replace this Section, placing it in front of Sections concerning “Disease Modeling and Microbial Infection”. We rearranged the description and added emphasis to make our opinion sound clearer.
Additionally, we checked the English in the manuscript thoroughly according to the advice. We hope it becomes better.
Finally, we are very grateful for the critical comments that the reviewer made, greatly appreciated his time-consuming work, but we note that these comments enabled us to change the present manuscript a lot and help us to understand the issue deeper.
